# Monitoring benthic plumes, sediment redeposition and seafloor imprints caused by deep-sea polymetallic nodule mining

Iason-Zois Gazis ●[1] ✉, Henko de Stigter ●[2], Jochen Mohrmann[1], Karl Heger ●[1], Melanie Diaz ●[2], Benjamin Gillard[3], Matthias Baeye[4], Mario E. Veloso-Alarcón[1], Kaveh Purkiani[1,5,6], Matthias Haeckel[1], Annemiek Vink ●[7], Laurenz Thomsen[3,8] & Jens Greinert ●[1,9]

A deep-sea (4500 m) trial of a pre-prototype polymetallic nodule collector with independent scientific monitoring revealed that a gravity current formed behind the collector channeled through steeper seafloor sections and traveled 500 m downslope. The prevailing bottom currents dominated sediment dispersion up to the end of the monitoring area at 4.5 km distance. The maximum suspended particle concentration recorded 50 m from mining lanes was up to four orders of magnitude higher than ambient values but decreased rapidly with increasing time, distance, and altitude. Most of the plume remained close to the seafloor, with the highest concentrations at 1 m monitoring altitude and reaching background concentrations at 50 m. Rapid particle flocculation was followed by fast and substantial sediment redeposition. A mm-scale photogrammetric seafloor reconstruction allowed quantitative estimates of the thickness of redeposited sediment next to mining lanes of ≈ 3 cm and a minimum erosional depth of 5 cm.

Almost 150 years after the first deep-sea polymetallic nodules (hereafter nodules) were recovered during the HMS Challenger expedition (1872–1876)[1], the industrial deep-sea trial of a tracked pre-prototype seafloor nodule collector vehicle (hereafter nodule collector) in the Clarion-Clipperton Zone (CCZ; eastern equatorial Pacific)[2] offered the opportunity to quantify the mobilisation, dispersal and redeposition of the benthic sediment plume (hereafter plume). Nodules are mineral concretions of millimetre to decimetre size formed over millions of years by precipitation of iron and manganese oxyhydroxides from seawater and sediment porewater around a nucleus[3]. They are embedded on the surface of ocean floor siliceous clays, typically between 3500 m and 6500 m water depth[3,4]. Due to their abundance and appreciable contents of copper, nickel, cobalt, and a range of rare earth elements, nodules have received commercial interest since the 1960s[4]. The surge of green, digital, and decarbonising technologies has increased this interest in the last two decades[4]. Mandated through the United Nations Convention on the Law of the Sea[5], the International Seabed Authority (ISA) is currently draughting the parts of the Mining Code[6] related to exploiting these resources. However, our knowledge of the monitoring technology needed, plume dispersion, sediment redeposition and likely impacts of deep-sea mining activities is still incomplete due to the recent birth of this industry and the complexity of acquiring data on the resilience of deep-sea life to such disturbances[7]. Valuable insights have been obtained from mining trials,

[1]GEOMAR Helmholtz Centre for Ocean Research Kiel, Kiel, Germany. [2]NIOZ Royal Netherlands Institute for Sea Research, Texel, The Netherlands. [3]Constructor University, Bremen, Germany. [4]Royal Belgian Institute of Natural Sciences, Brussels, Belgium. [5]MARUM Center for Marine Environmental Sciences and Faculty of Geosciences, University of Bremen, Bremen, Germany. [6]Now at Federal Maritime and Hydrographic Agency, Hamburg, Germany. [7]Federal Institute for Geosciences and Natural Resources, Hannover, Germany. [8]Now at the University of Gothenburg, Gothenburg, Sweden. [9]Institute of Geosciences, Christian-Albrecht University of Kiel, Kiel, Germany. ✉e-mail: igazis@geomar.de

and benthic impact experiments carried out since the 1970s, although those disturbances were of a different nature, spatial scale and intensity than those expected from industrial mining technologies[8–16].

Herein, the most detailed, up-to-date in situ view of deep-sea plume monitoring using hydroacoustic and optic sensors is presented. The Patania II nodule collector of the Belgium-sponsored ISA contractor Global Sea Mineral Resources NV (GSR) is smaller in size and weight than an industrial-scale prototype collector and was tested without a riser system[2,17]. However, it does integrate caterpillar tracks for locomotion and hydraulic suction for nodule collection, a technological scheme widely considered feasible for future mining activities[18]. The 4 m wide hydraulic nodule collector head uses water jets to lift the nodules (Coandă Effect)[17], thereby also mobilising the topmost bioactive layer of the seafloor with its epi- and endofauna as well as its microbial communities[19]. The resulting mixture of nodules, sediment and water passes through a separator, which diverts the nodules into the collector bucket, while the sediment-laden water passes through the diffuser located at the rear side and is discharged as a plume at 2.5–3.2 m above the seafloor[17,20]. Together with sediment stirred up by the locomotion on the seafloor, the initial input to the plume during the trial was estimated to be $12 \pm 3\,kg\,s^{-1}$ of sediment[20].

Independent scientific monitoring of the plume dispersion and benthic ecosystem disturbance was conducted in the framework of the third-party-funded MiningImpact 2 project[21] of the European Joint Programming Initiative entitled Healthy and Productive Seas and Oceans[22] using the MV Island Pride[2]. More than ten years of oceanographic observations, detailed analyses of the seabed sediment characteristics and spatial distribution of nodules, insights into the aggregation potential of suspended sediment from laboratory and small-scale deep-sea disturbance experiments, as well as numerical models of plume behaviour, provided the basis for the seafloor and plume monitoring strategy applied during the mining trial[12,13,16,19,23–29]. Many of the monitoring platforms and sensor settings had already been tested during a dredge disturbance experiment in the CCZ[12]. In total, 44 oceanographic sensors were mounted on 23 stationary seafloor platforms[2] distributed at various distances (50–1800 m) from the collector impact site (hereafter impact site), including a PartiCam[29] sensor for in situ particle size monitoring. Two Remotely Operated Vehicles (ROVs) were used to visually observe the plume and deploy/ recover the multi-sensor platforms[2]. An Autonomous Underwater Vehicle (AUV) acquired multibeam echosounder (MBES), side-scan sonar (SSS), and seafloor image data, as well as suspended particulate matter (SPM) concentration (hereafter concentration) at different altitudes (5, 10, 30 and 50 m) and distances (up to 5 km) from the impact site[2]. The optical backscatter sensors (OBS) for measuring concentration ($mg\,L^{-1}$) were calibrated to reference suspensions prepared from CCZ surface sediments (see Methods). All OBS and acoustic doppler current profilers (ADCPs) measured ambient concentration and background acoustic backscatter intensity before the trial.

## Results

### Mining imprints on the seafloor

On 19 April 2021, the nodule collector was lowered to the seafloor and was remotely operated for 41.33 h, covering a seafloor distance of 21.37 km². It picked up nodules along 171 lanes, each 50 m long and 4 m wide, resulting in a total mined area of 0.034 km². The displaced nodules ($\approx 660$ t) were deposited on the seafloor as piles at the end of each driven lane ("light bulb" turn manoeuvre; Figs. 1–2 and Supplementary Fig. S1). The lanes were distributed over three strips (first strip: 55 lanes, second strip: 31 lanes, third strip: 85 lanes; Fig. 1). The first and third strips were driven from SW to NE, whilst the second strip was driven from NE to SW. The two northern strips were located on a gentle NE-sloping seafloor (0°–3°), whereas the southern strip ended towards the NE in a steeper (3°–6.5°) slump scar-like structure sloping

down into a wider N-S oriented enclosed basin (Fig. 3). The SSS mosaic shows that $\approx 0.08$ km² of the seafloor was directly disturbed by the nodule collector, including the turns (Fig. 1). The mined lanes are cleared from nodules and eroded by the water jets. Based on the Digital Elevation Model (DEM) derived from seafloor images acquired 88–110 h after the trial within the impact site, an erosional depth of $\approx 5$ cm was measured. However, this depth is measured after substantial sediment redeposition occurred (see Sediment redeposition). The caterpillar tracks created an additional small-scale topography due to sideways sediment extrusion (Fig. 2 and Supplementary Fig. S2).

### Spatial distribution of the plume

At 50 m distance from the edge of the impact site, the maximum concentration recorded was $264\,mg\,L^{-1}$ at 1 m altitude (NIOZ_PFM-06; Fig. 3). This was four orders of magnitude above the measured background level of $0.02–0.03\,mg\,L^{-1}$, but returned to ambient values after 14 h (Supplementary Figs. S2, S3). At the flattest parts within the impact site (Fig. 1), the ROV recorded a low-lying plume with a sharp front distinct from the ambient water and headed perpendicular to mining lanes. Although a gentle oscillation was observed, the net movement was downslope in the form of a gravity current (Fig. 3).

NE of the impact site, at a distance of $\approx 200$ m downslope, a sharp increase in the concentration at 1 m altitude was recorded at NIOZ_PFM-07, which lasted for 2.3 h before returning to background values (Supplementary Fig. S2). The plume arrival was marked by a distinct increase in acoustic backscatter intensity, current speed and current direction change from S to E based on ADCPs at NIOZ_PFM-07 and GMR_PFM-39 (Fig. 3 and Supplementary Figs. S4, S5). A progressive vector plot at 1.2 m altitude revealed a 380 m eastward displacement during the passage of the gravity current (Fig. 3). The gravity current created sediment ripples with an NE–E current direction (Figs. 3, 4). The presence of sediment ripples could imply the mobilisation of additional surface sediments due to erosion[30]. This would, however, demand current velocities $> 30\,cm\,s^{-1}$ as determined during experiments on sediment resuspension in a large seawater flume using original sediments from the CCZ or as determined for deep-sea sediments of similar particle composition[29,31,32].

We consider a more likely explanation is that the massive fallout of aggregated sediments within the gravity flow[33] (see Particle flocculation) combined with the unidirectional currents in steeper seafloor parts resulted in mud ripples generated by large non-cohesive plume aggregates. They behave hydraulically equivalent to fine sands[34,35]. Results from laboratory studies of aggregated plume sediments from the study site[23] using the approach of Thomsen & Gust[31] revealed that bedload transport generated ripples began to form at a critical shear stress of $0.04\,N\,m^{-2}$ and resuspension of recently settled plume sediments starts at $0.1\,N\,m^{-2}$ (see Online Video Content). Multiple turbidity currents originating from multiple lanes could enhance the shape and dimensions of ripples. Larger sediment ripples were mapped at the NE end of the first mining strip (downslope direction), where the gravity current focused into the steeper NE running valley (Fig. 4). Here, the nearest OBS (JUB_PFM-01) recorded the second-highest concentration ($169\,mg\,L^{-1}$; Fig. 3), while the ADCP data at NIOZ_PFM-03 shows a change in current direction from S to E-NE at the same time (Supplementary Fig. S6). No sediment ripples were found upslope of the mining strips, but nodules were entirely covered by redeposited sediment (Fig. 2).

SE of the impact site, at 500 m distance downslope, the 2 MHz ADCP at NIOZ_PFM-04 recorded a change in current direction from SE to E, but not in magnitude (Supplementary Fig. S7). This implies that the gravity current had weakened at that distance. However, the comparison with the ADCP at NIOZ_PFM-07 might be flawed by the relatively low sampling interval of 5 min[2] and the absence of video data during the time of ADCP measurements that could ensure that the ADCPs measured the same part (e.g., head) of the gravity current.

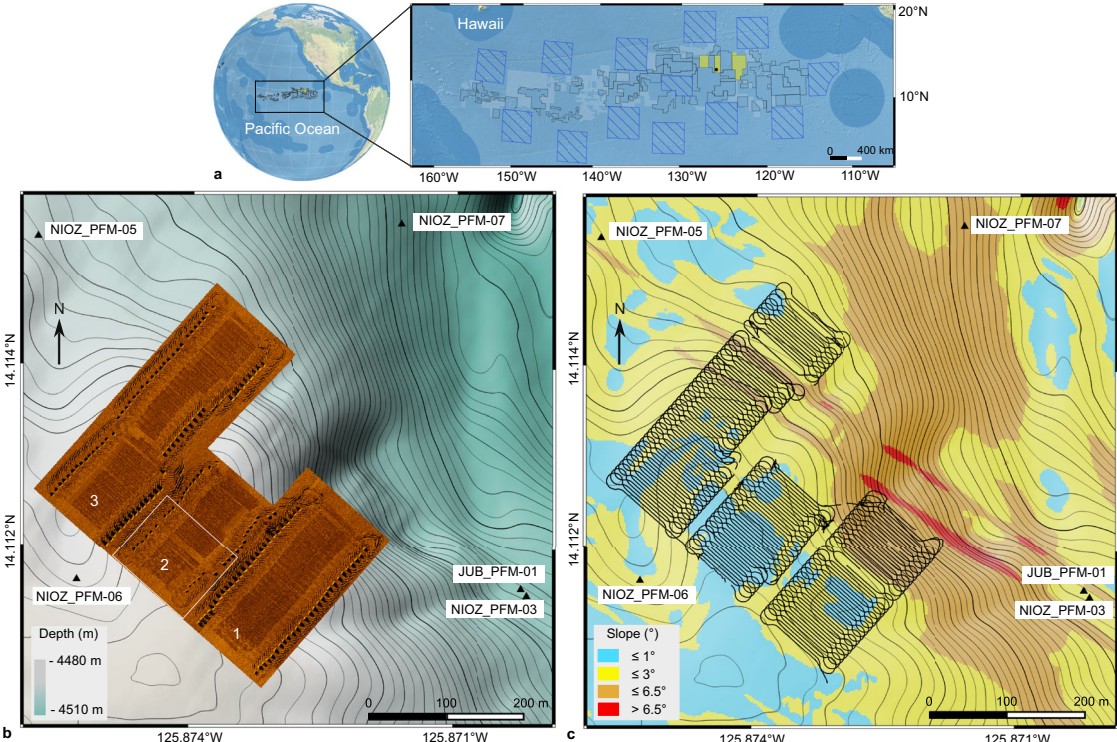

**Fig. 1 | The three mining lanes and the seafloor morphology of the nodule impact site. a** Global map[85] and the Clarion-Clipperton Zone (CCZ), between Central America and Hawaii, where most of the current polymetallic nodule exploration contract areas are located (black polygons)[86]. The Global Sea Mineral Resources NV exploration contract area, which is divided into three blocks, is marked in yellow. The red dot shows the location of the mining trial described in this study. The blue hashed squares represent Areas of Particular Environmental Interest (i.e., no mining areas), and the light blue blocks are Reserved Areas (i.e., areas reserved for developing nations)[86]. Exclusive Economic Zones[87] are shown in darker blue. **b** Side-scan sonar map showing the nodule collector mining lanes and strips superimposed on the bathymetry (0.5 m isobath contours) and the locations of nearby sensor platforms. The mining lanes across the three parallel strips are visible due to their lower backscatter intensity related to nodule removal. The white box inside the second strip indicates part of the orthophoto-mosaic area presented in Fig. 2. **c** Slope distribution map of the impact site. Multibeam echosounder residual artifacts cause the NW-SE linear trending seafloor parts with slopes > 6.5° (red coloration).

Nevertheless, this was the most distant location where such a change in the current direction was observed, implying the end of the active phase of sediment transportation as a gravity current. A 6 m high sill with slopes of up to 3° along the plume's path blocked the further eastward propagation, and the plume spread laterally (up to 690 m) around NIOZ_PFM-04. Here, the AUV mapped the lateral spreading of the plume's lower part (< 5 m altitude) using MBES Water-Column Imaging (WCI) data (Figs. 3, 5).

Beyond this distance, the plume drifted passively with the ambient bottom currents in the SE to S direction (Supplementary Figs. S8–S10). The OBS mounted on the most remote platform at ≈ 1800 m from the impact site (NIOZ_PFM-01; Fig. 3) recorded a maximum concentration of 3.9 mg L⁻¹ at 1 m altitude (Fig. 3 and Supplementary Fig. S2). This concentration is three orders of magnitude less than that recorded 50 m from the impact site but is still 100-fold higher than the background concentrations. The plot of the maximum concentrations from OBS located SE of the impact site showed a power-law decay in plume concentration with increasing distance and time at 1 m altitude (Fig. 6). This was visually confirmed during an ROV dive, where the top of the plume was followed. This approach worked well inside and near the impact site where the plume was still quite dense, but towards the SE at ≈ 1700 m distance, it became increasingly difficult to distinguish the top of the plume from overlying clear water.

With the AUV flying at 5 m and 10 m altitudes, it was possible to track the plume up to 4.5 km S of the impact site 35 h after the start of the trial. At that distance, the concentration was 0.1 mg L⁻¹. The two AUV tracks north of the impact site only recorded background-level values, confirming the SE-S passive advection of the plume by bottom currents (Fig. 3 and Supplementary Fig. S11). The plume was detected over an area of ≈ 9 km².

N and NW of the impact site, at a distance of ≈ 200 m upslope and upstream, the lowest maximum concentrations (0.05–0.14 mg L⁻¹) and backscatter values (34–49 dB) were recorded (Fig. 3 and Supplementary Figs. S2, S12, S13).

With regards to the vertical spreading of the plume, the multi-altitude (5, 10, 30, and 50 m) passes of the AUV at 500 m distance SE from the impact site also showed a power-law decrease in maximum concentrations for the survey period (Fig. 6). The upward-looking ADCPs showed a rapid decrease in backscatter intensity with increasing altitude (Fig. 7 and Supplementary Figs. 1,3–6,10). Moreover, they revealed that while the lower (< 5 m altitude) and denser (up to 35 mg L⁻¹ at 1 m altitude) part of the plume moved towards the E, the upper part (with a maximum concentration of 3.2 mg L⁻¹ at 10 m altitude) moved towards SE like the prevailing bottom currents (Supplementary Fig. S7). The 300 kHz ADCPs recorded the plume at altitudes > 30 m at 1800 m SE from the impact site. Together with the AUV OBS records of ≈ 0.07 and 0.06 mg L⁻¹ at 30 and 50 m, respectively (500 m from the impact site; Fig. 3 and Supplementary Fig. S11), these data show that suspended particles can reach greater heights with distance and time but in low concentration.

The plume evolution has been compiled in Digital Earth Viewer[36], a web-based 4D visualisation software that allows easy navigation through time and space. Moreover, a time-lapse video from a still camera (GMR_CAM-37) next to GMR_PFM-43[2] shows the succession of the first two plume episodes at < 1 m altitude (see Online Video Content).

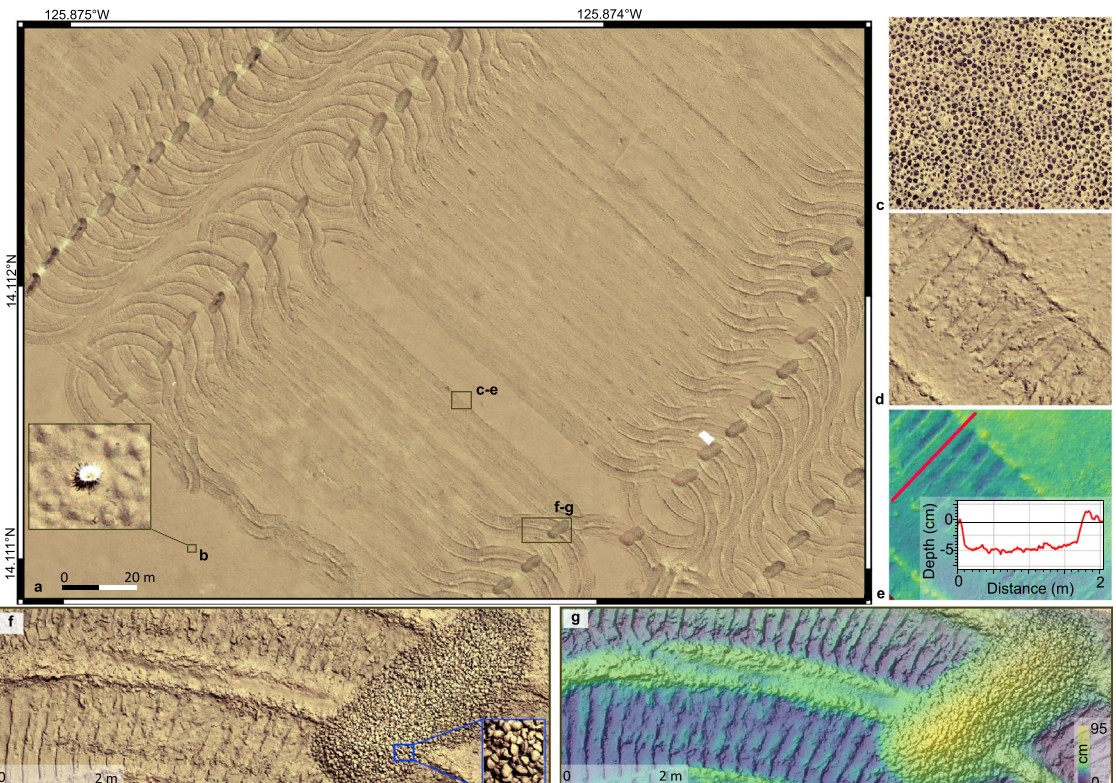

**Fig. 2 | Orthophoto-mosaics obtained at a depth of 4500 m show the mining imprints on the seafloor. a** Orthophoto-mosaic with 2 mm × 2 mm pixel resolution from part of the second mining strip (Fig. 1b), showing the nodule collector imprints on the seafloor as mining lanes, the "light bulb" turns, and nodule piles. **b** A sea anemone (*Actiniaria*)[88], a benthic animal capable of cleaning itself from sediment[89], is shown as an example of the resolution achieved. **c, d** Orthophoto-mosaic of the same seafloor part before (**c**) and after (**d**) the mining trial. **e** Image-derived digital elevation model (DEM) of the seafloor part shown in (**d**). A depth profile across the caterpillar track (red line) shows the erosion depth caused by the nodule collector and the humps on each side of the track due to sideways sediment extrusion under the caterpillar tracks. The zero-depth refers to the unmined seafloor between the two caterpillar tracks along the same mining lane. The maximum depth shown has been formed after the sediment redeposition and consequently could be smaller than the true erosion depth. **f** Orthophoto-mosaic of the track of the nodule collector while performing the light bulb turning manoeuvre at the end of a lane and the deposited nodule pile (≈ L 5 m × W 2 m × H 0.9 m). The sideways sediment extrusion under the caterpillar tracks is larger along the "light bulb" turn manoeuvres. **g** Image-derived DEM of the same area with a pixel resolution of 5 mm × 5 mm. A three-dimensional orthophoto-mosaic of this area is also shown in Supplementary Fig. S1.

## Particle flocculation

The fact that the plume can be observed with ADCPs at 300 kHz with low sensitivity for primary particle size (median ≈ 0.02 mm)[37] indicates that at least part of the mobilised fine-grained sediment must be present in an aggregated form that falls within the detection window of these devices. The PartiCam (JUB_PFM-01) showed that whereas the median particle size ($D_{50}$) of the top 5 cm of the seafloor sediment is 0.012 mm[17], the plume $D_{50}$ increased up to 0.147 mm during the second and third plume episodes (Fig. 8). Throughout the trial, which lasted longer than the minimum time needed to form flocs with CCZ sediment in the lab[23,38], the first ($D_{25}$) and third quantile ($D_{75}$) ranged from 0.061 mm to 0.313 mm (Fig. 8, Table S1 and Supplementary Fig. S14). The largest particles were detected during the third plume episode when the plume was produced furthest away from the camera and exhibited the longest aggregation time (Fig. 8, Supplementary Table S1 and Supplementary Fig. S14). Although the settling velocity was not measured in situ, laboratory experiments with similar-sized aggregated plume particles from the study area revealed settling rates of ≈ 100 m per day[23,39]. Such rates resulted in short residence times within a 2-3 m high plume of 30 – 45 min, with more than 80 % of all aggregated particles settled by then[39]. If these particles settle within the first 500 m away from the nodule collector, the corresponding flow velocity of the gravity current would be ≈ 0.15 m s$^{-1}$ for the $D_{50}$ and ≈ 0.2 m s$^{-1}$ for larger particles $D_{50}$ - $D_{75}$. These values are close to the

discharge buoyancy velocity of ≈ 0.21 m s$^{-1}$ mentioned in the near-field study[20]. They also agree with the in situ observations from the OBS sensor at NIOZ_PFM-06, which registered the plume for the first time on 19.04.2021 at 19:25:00. Assuming an average and constant speed of ≈ 0.2 m s$^{-1}$ for a period of 600 s, the nodule collector should have been ≈ 120 m away from the platform at the time of sediment discharge. Plotting the nodule collector track positions shows that on 19.04.2021 at 19:15:00, the collector was 124 m away from NIOZ_PFM-06 on a flat seafloor.

## Sediment transport and redeposition

The seafloor orthophoto-mosaic and ROV footage that followed the trial showed that the unmined seafloor within the impact site and in the close vicinity (< 100 m) was completely draped by redeposited sediment so that nodules did not appear as dark objects anymore, i.e., sediment blanketing occurred (Fig. 9 and Supplementary Fig. S15). As the height of the nodules above the sediment surface at this site is 1–3 cm[17,40], the sediment redeposition was at least 3 cm in these locations (Fig. 9 and Supplementary Fig. S15). The redeposited sediment smoothed out the convex shape of nodules and filled the gaps between neighbouring nodules, resulting in decreased seafloor ruggedness (Fig. 9). Throughout the impact site's first 500 m SE downslope (along the gravity current propagation path), AUV and ROV seafloor images also showed substantial sediment blanketing

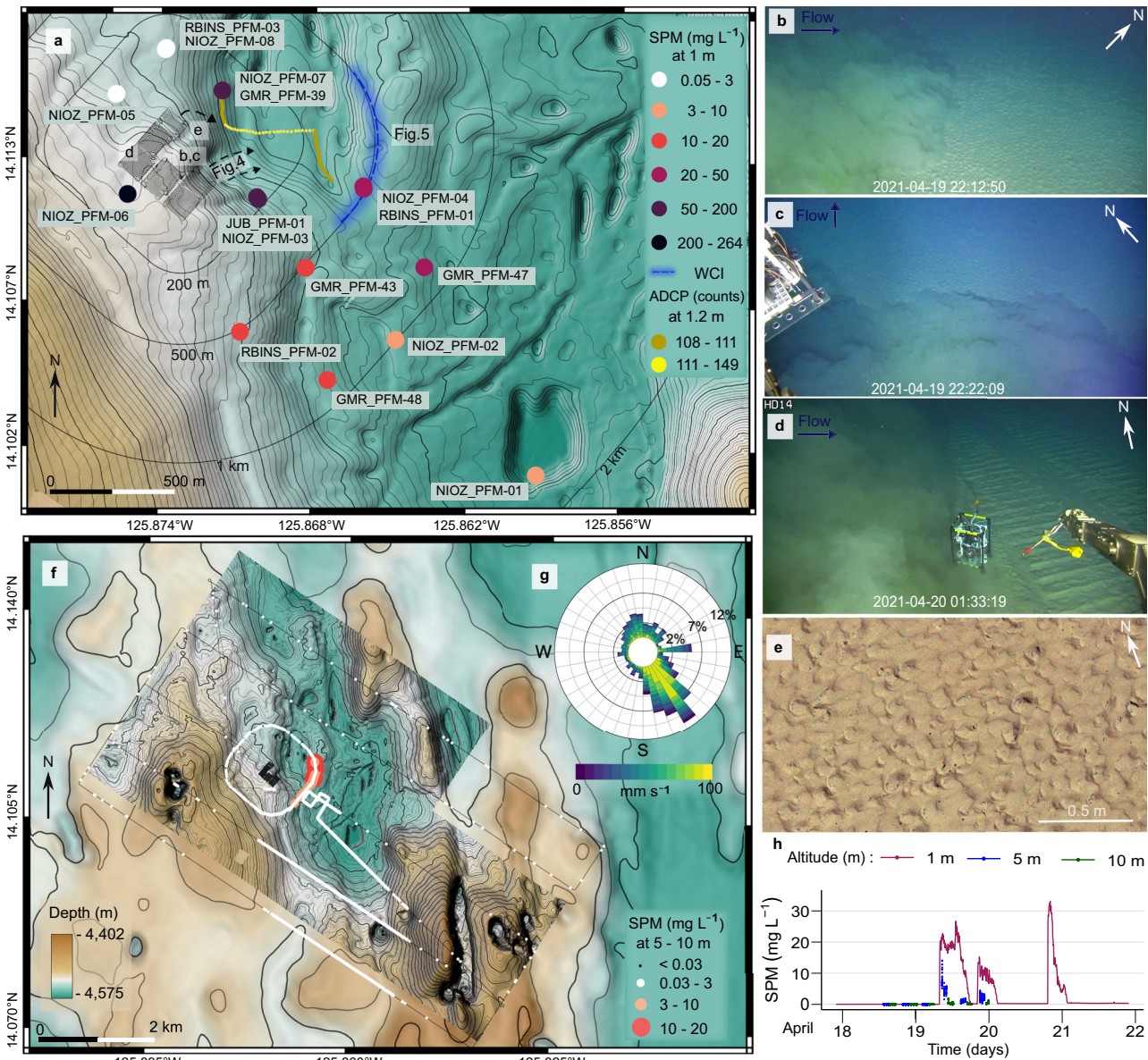

**Fig. 3 | Spatiotemporal distribution of the benthic sediment plume. a** Maximum concentration registered by optical backscatter sensors (OBS) at 1 m altitude. Time-series graphs of concentration for all OBS are provided in Supplementary Fig. S2. The acoustic doppler current profiler (ADCP) trajectory plot of the NIOZ_PFM_07 platform showed an eastward displacement and increased backscatter intensity when the gravity current passed. The extent of the multibeam echosounder water-column imaging (MBES WCI) data agrees with the OBS data from the autonomous underwater vehicle (AUV) (**e**). **b**, **c** Video frame grabs inside the nodule impact site show the edge of the benthic sediment plume (plume) being distinct from the ambient seawater. Limited vertical mixing is also observed as the altitude of the remotely operated vehicle (ROV) is ≈4.8 m. Parts of the ROV footage are available online (see Online Video Content). **d** ROV image[2] showing the plume heading

perpendicular to mining lanes. The sensor shown (Microprofiler)[2] is ≈80 cm in height and 60 cm in width. **e** Sediment ripples at the end of the third mining strip. **f** The AUV recorded the spatial extent of the plume during a ca. 54 h survey conducted parallel to the mining trial and continued after its termination. Scattered individual measurements above 0.03 mg L⁻¹ correspond to noise, e.g., from fish inside the sensor's view. The background map is the AUV-based MBES bathymetry of the monitored area superimposed on the ship-based MBES bathymetry[70]. **g** The Rose diagram at NIOZ_PFM-04 represents the current direction and speed during the nodule collector trial. The eastward spike is related to the gravity current. **h** Concentration recorded at 1 m, 5 m, and 10 m altitude at NIOZ_PFM-04. The measurements at 5 m and 10 m altitude had to last shorter to facilitate the far-field survey. The locations of Figs. 4 and 5 are also noted.

(Fig. 10 and Supplementary Figs. S15–S17). A simple estimation of the sediment mass transported at 500 m SE from the impact site was attempted using the MBES WCI data and the OBS data from NIOZ_PFM-04 and RBIS_PFM-01 (see "Methods"). During a two-hour mining activity in the first strip, 11.2–12.3 % of discharged sediment mass passed through the lower 2 m and an additional 0.81–1.02 % of sediment mass was estimated to pass at 5–6 m altitude (Supplementary Fig. S18). This estimation sets a lower limit, with the actual

percentage expected to be higher as the locations of platforms are on the southern edge along the recorded MBES WCI data, where lower concentrations were observed (Fig. 3). Moreover, the OBS at those platforms measured concentration at 1 m altitude. However, a portion of the plume had already ascended to higher altitudes (see Discussion).

At greater distances from the source, computer-based quantitative analysis of seafloor images (see "Methods") showed that the

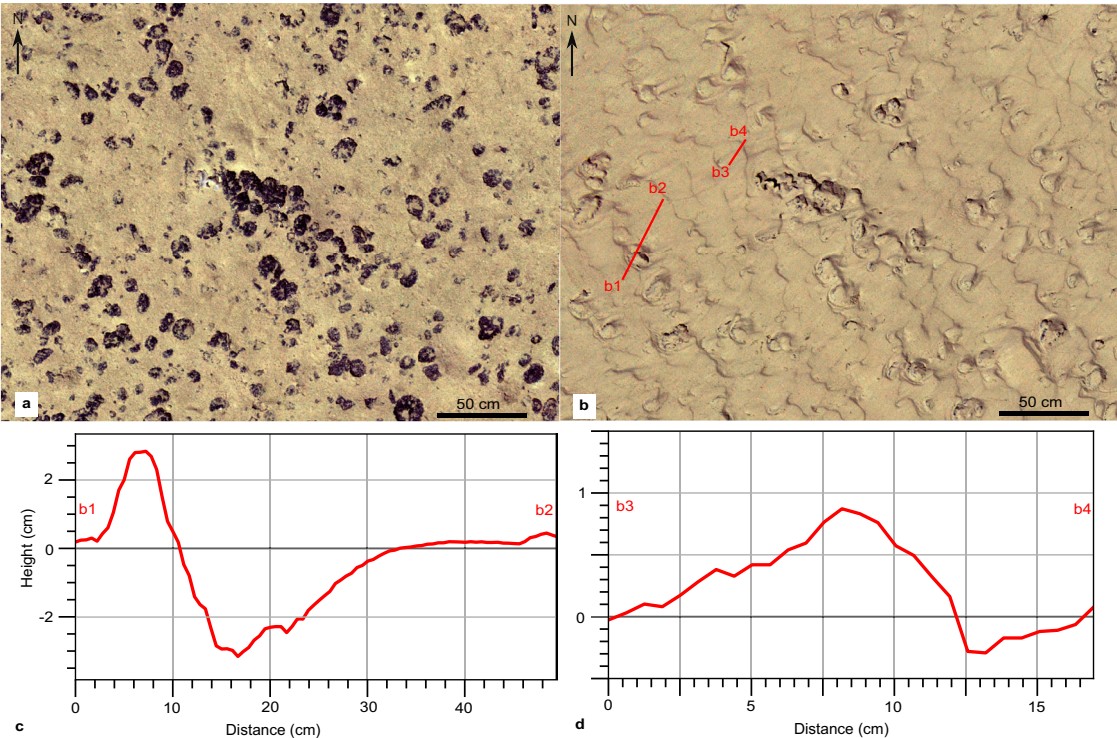

**Fig. 4 | Formation of ripples downstream and downslope from the first mining strip. a** Orthophoto-mosaic of seafloor part downslope from the end of the first mining strip before the mining trial. **b** The same seafloor part after the mining trial. The formation of ripples heading towards the East (downslope direction) is depicted. **c, d** Topographic profiles based on the image-derived digital elevation model (DEM) after the mining trial. The depth of sediment erosion behind the nodule (**c**) and the ripple's lee side are shown (**d**). The ripple geometry (stoss side, crest, and lee side) is outlined (**d**). The ripple crests have a shape sinuous to catenary, which is typical for strong flow over cohesive sediments[90].

detectable % seafloor nodule coverage ranged between 20 % and 30 % before the mining trial. After the passage of the plume, it dropped to 2–10 % (Fig. 10 and Supplementary Fig. S17). We observed three types of sediment redeposition: thick, medium, and faint coverage (see Methods for classification scheme). The sediment redeposition progressively faded towards the SE direction, where coverage could no longer be visually distinguished on the seafloor images at 1800 m. The faint blanketing is more extended in a southward direction (Fig. 10), whilst no sediment blanketing was observed along the two northeastern AUV lines, agreeing with the AUV OBS data (Fig. 3).

Within the limits of space and time during our monitoring survey, sediment redeposition was detected over an area of ≈ 6 km² (Fig. 10). Estimating conservatively the total volume of mobilised sediment to be 1700 m³ (8500 m of collector lanes, 4 m collector width, 0.05 m average erosion depth, without considering sediment suspension around the collector heads and caterpillar tracks), and assuming that all mobilised sediment would settle out over ≈ 6 km², an average coverage thickness of ≈ 0.2 mm is hypothesised (assuming sediment density equal to that of the undisturbed top 0–5 cm layer)[17]. Knowing that a small fraction of the mobilised sediment likely travelled out of the monitored area and that deposition within the monitored area was focused close to the impact site (< 500 m), this 0.2 mm of coverage represents the upper limit in distal parts, which we expect to be even smaller.

## Discussion

Our results demonstrate that the sediments mobilised by the nodule collector generated a gravity current that propagated perpendicular to the mining lanes to distances of up to 500 m downslope from the impact site. Up to this distance, the seafloor slope determined the E-ward direction of the gravity current. The ambient SE-directed crossflow did not deflect the gravity current from its downslope trajectory within the first 500 m. At the base of the slope, a stretch of flat seafloor (< 1°) bounded towards the East by a sill forming an enclosed basin. The sill blocked the propagation of the gravity current further to the East, and the plume started spreading laterally (Fig. 5). Such flow refraction/diversion of gravity currents has also been shown in other studies[41,42]. From there onward, ambient currents determined the plume's direction towards SE, with finer particles still in suspension dominating the plume composition. The gravity current travelled downslope for longer distances than reported from initial tests of the same nodule collector on flat terrain (< 1°)[20]. As our results indicate that seafloor morphology plays an essential role in gravity current propagation, increasing the area of thick sediment redeposition, we suggest that the AUV-scale mapping would be necessary for future monitoring and plume modelling, while the mining could be prioritised to flat areas limiting the spatial extent of thick sediment redeposition.

The fast sediment redeposition within a limited spatial extent was supported by particle flocculation, observed in situ, corroborating previous lab[23,33,38] and field studies with nodule collectors[8,43,44], deep-sea epi-benthic sledges[10–13,16], ROV-induced plumes[45], trawling gear in muddy sediments[46,47] and redeposition models derived from those tests[11,28,48]. Similarly, empirical power-law shape functions have been reported to describe the sediment concentration decay in low-concentration plumes under currents[49,50], deep-sea hydrothermal plumes[51] and plumes generated by towed fishing gear[46,47]. This power trend, which expresses the balance between the settling of the SPM (due to gravity) and upward lifting (due to turbulence), can be described by the generalised advection-diffusion equation[52], if mixing and bottom current advection dominates the plume (which is what occurred here in larger distances and altitudes). At shorter distances

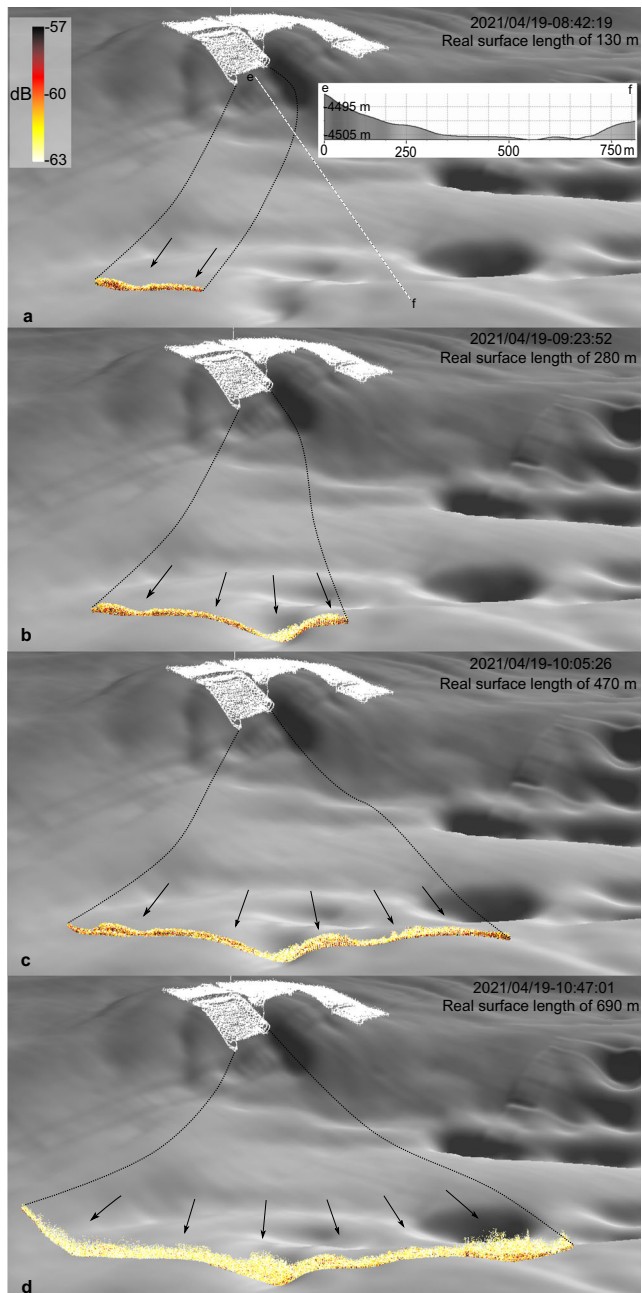

**Fig. 5 | Multibeam echosounder water-column backscatter data 500 m SE of the impact site. a–d** Time-series plot of the multibeam echosounder (MBES) water-column backscatter data 500 m SE from the impact site, showing the evolution of the benthic sediment plume's lower part (< 5 m), corresponding to the first mining strip. The seafloor topography blocked further eastward propagation, and the plume spread laterally. The time (in UTC) corresponds to the start of each MBES file. The seafloor vertical exaggeration is × 9.

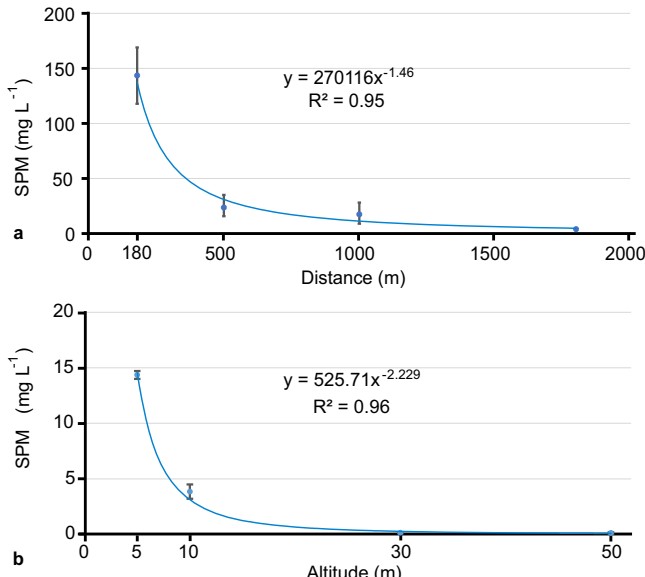

**Fig. 6 | Power-law trend line as functions of distance and altitude for maximum concentrations of suspended particulate matter observed. a** Maximum suspended particulate matter (SPM) concentration recorded by optical backscatter sensors (OBS) at successive distances (from 180 m to 1800 m) and times (19 to 22 April 2021) in the SE direction from the impact site. The maximum values from the OBS on the following platforms were used: JUB_PFM-01 and NIOZ_PFM-03 at 180 m ($n = 3$), NIOZ_PFM-04, RBINS_PFM-01, GMR_PFM-43 and RBINS_PFM-02 at 500 m ($n = 4$), GMR_PFM-47, NIOZ_PFM-02 and GMR_PFM-48 at 1 km ($n = 3$), and NIOZ_PFM-01 at 1.8 km ($n = 1$). The sensors' locations are given in Fig. 3, and their time series data in Supplementary Fig. S2. When more than one OBS were available, the averaged maximum value was calculated and plotted (blue circle). The vertical bars represent the highest and lowest maximum values measured; this variability is due to the different locations across the longitudinal axis at the same distance from the impact site. **b** The averaged maximum SPM concentration recorded by the two ($n = 2$) OBS mounted on the autonomous underwater vehicle (AUV) at successive altitudes (from 5 m to 50 m) and times (19 to 20 April 2021) while circling at a 500 m distance around the impact site. As two OBS were available at all altitudes ($n = 2$), the average maximum value was calculated and plotted (blue circle). The vertical bars represent the highest and lowest maximum values measured; this variability is due to a) different mounted locations on AUV and looking directions (forward and upward). For a detailed comparison, see Supplementary Fig. S11.

investigated. Here, it was not possible to establish a clear relationship between the concentration gradient and the total percentage of sediment mass passing by in the lower 5 m above the seafloor, as only one measurement at 1 m altitude exists. Our empirical data cannot be extrapolated and applied generally over larger scales or for other trials with different initial conditions (e.g., discharge rate) or other nodule collectors, different sediment size distribution, seafloor morphology and bottom currents without proper understanding of the physical processes contributing to the observed dispersion pattern. However, it does provide a detailed representation of the investigated trial.

The undertaken AUV-mounted MBES survey proved that the lower plume cross-section can be observed if the AUV flies across the plume axis during a low altitude survey and the plume is still highly concentrated, which shows the potential of MBES WCI data to estimate concentrations and grain size, at least for larger aggregates[56–59]. The MBES frequency and settings used[2] could not detect the plume during the far-field AUV survey, where the concentration was ≈ 0.1 mg L$^{-1}$ at 5 m altitude and detectable by the OBS on the AUV (Fig. 3). The combined use of two (or more) AUVs that ensonify the plume along and across the spreading axes using higher MBES frequencies (700–1400 kHz) and seafloor platforms with forward-looking[59] or 360° rotating MBES[60], would close the gap towards a 3D mapping of the

(here < 200 m) and lower altitudes (here < 5 m), the plume shows little mixing, and it is dominated by the near-field gravity current dynamics generated by the mining vehicle[20,53,54]. In future surveys, a denser OBS and ADCP grid closer to the source and along the gravity current propagation path is favourable to depict the concentration decay at short distances and capture the ignition, auto suspension and dissipation phase of the gravity flow[55]. The required time for the mean floc size to form and break up again and its effect on settling velocity within the gravity current and at greater distances should also be

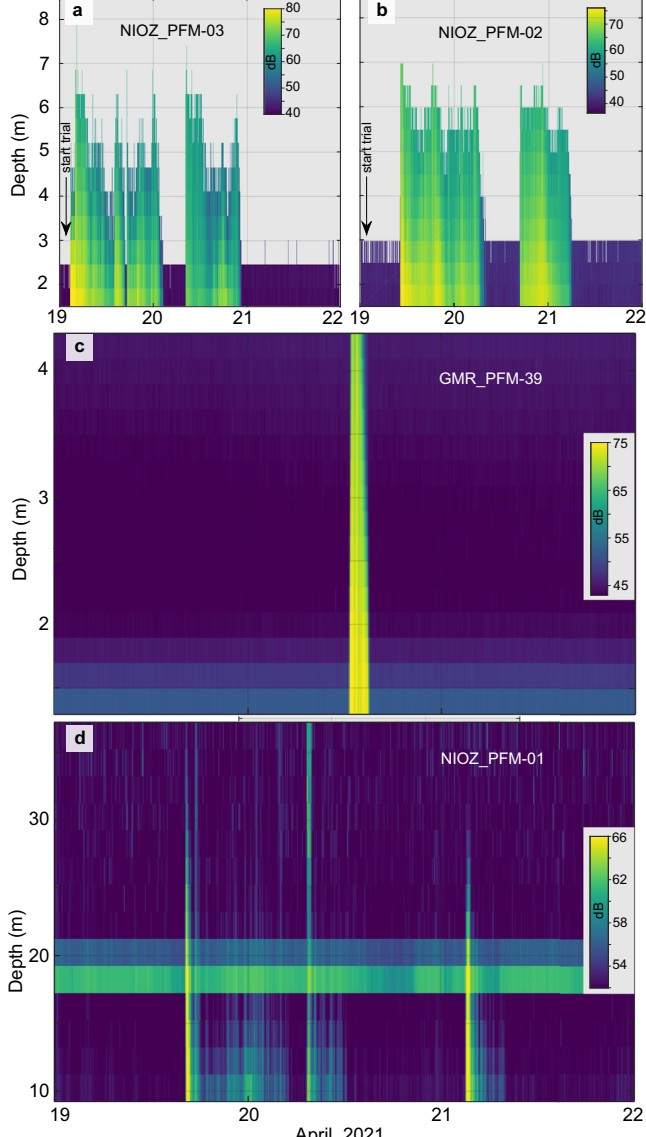

**Fig. 7 | Acoustic doppler current profilers recorded the benthic sediment plume. a, b** Water-column backscatter data from two upward-looking acoustic doppler profilers (ADCPs) at 2 MHz located at different distances SE from the impact site. The three benthic sediment plume (plume) episodes are visible due to the sharp increase in backscatter intensity. A ≈ 10 dB decrease in maximum backscatter intensity was observed between the two ADCPs, indicating fewer particles at greater distances. **c** An upward-looking ADCP at 1.2 MHz shows a sharp increase in backscatter intensity when the gravity current passes. **d** An upward-looking ADCP at 300 kHz, almost 2 km SE from the impact site, shows the three plume episodes having a larger height. A flotation body on the platform causes the background noise between 18–22 m above the seafloor. All backscatter echo values are normalised. Each ADCP could capture only part of the plume's vertical extent as different frequencies and settings were used[2].

plume volume at least for the near-field monitoring. The fate of the plume, as it spreads over longer distances and time spans, and the thickness of the redeposited sediment can currently only be derived from 4D numerical models. Improved by in situ data such as those presented here and accounting for particle flocculation[23] and seafloor morphology, such models can provide reliable forecasts under different initial conditions and take account of sporadic environmental events, such as mesoscale eddies reaching the seafloor[61–63]. In this regard, the deployment of long-term seafloor observatories at successive distances from the impact site could enrich our knowledge of

far-field plume dispersion, as an older mining trial with a towed nodule collector and different monitoring technology detected the plume in distances up to 10 km downstream[8].

Whereas observations made at short range from the nodule collector showed that the plume stayed mostly below 2 m altitude[20], our results demonstrate that it expanded in height, albeit at a much lower concentration, while spreading from the source. Similar observations have been reported during deep-sea mining pre-prototype collector tests in shallow water[43] and benthic disturbance experiments[45]. In addition, the maximum concentration recorded here is higher than reported from initial testing of the same nodule collector[20]. These differences show that more in situ trials must be closely monitored, particularly with larger nodule collectors or nodule collectors that use different technology. It is important to note that the larger size and longer operation time of a full-scale nodule collector, increasing from a few days to multiple years, will - in comparison with this trial - result in a much larger seafloor mining area (≈ 200 km² per year) and volume of resuspended and redeposited sediment over the lifetime of the mine[18]. The method employed for mining, whether along extensive routes or focused on smaller areas, as well as the seafloor slope, would also affect plume dispersion.

In addition, the sediment in suspension around the collector heads and caterpillar tracks should be considered. This resuspended sediment is not entrained into the nodule collector, having different characteristics (e.g., sediment size distribution) and behaviour than the discharge plume. It contributes to the total plume concentration within the impact site, but it is unclear if it is integrated into the gravity current and far-field plume transportation. Moreover, the merging of multiple gravity currents generated by the mining vehicle(s) is a field that should be investigated in detail[64]. Both could be of interest if one or more nodule collectors operate continuously for weeks or months over the seafloor. The erosion depth under the caterpillar tracks is expected to be of similar scale as long as the mass per surface contact area remains similar to the pre-prototype used here and the sediment characteristics (e.g., shear strength) are the same. Using image-derived DEMs to calculate the erosion depth provides good spatial coverage, but it only detects the minimum depth as it cannot account for redeposited sediment that has already settled before the image data acquisition. The slurry produced by onboard processing of collected nodules and discharged via a riser system near the seafloor will increase the SPM in the benthic environment with potentially cumulative effects. However, release close to the seabed may help limit the spreading of the discharge plume and reduce the environmental footprint, thus being preferred over midwater release.

This data-driven study tested existing sensors and technologies for deep-sea plume monitoring as an analogue to the equipment and approaches that we think will most likely be used in future industrial mining scenarios. The results show that the available sensor technology and the combination of fixed and moving platforms can quantify the evolution of plume concentration in space and time, while the image-derived 3D reconstruction of the impact site yielded a resolution down to mm scale that was not available until now. Despite these advancements, two main issues should be addressed in future studies.

Firstly, the post-processing, analysis, collation, visualisation, and interpretation of terabytes of data from different sensors is presently a task that demands computational power and time during onshore work. The current practice does not allow effective decision-making informed by real-time data ingest during future mining activities. Seagoing high-performance computers for image, MBES, SSS analysis, and adaptive spatiotemporal plume modelling (similar to weather forecasts) should become part of the monitoring equipment on mining vessels, an approach already introduced on research vessels[65]. Using such data and numerical models linked to 4D visualisation capabilities that enable extracting, exploring, and visualising quantitative and semantic information in near-real time (digital twin) would allow for

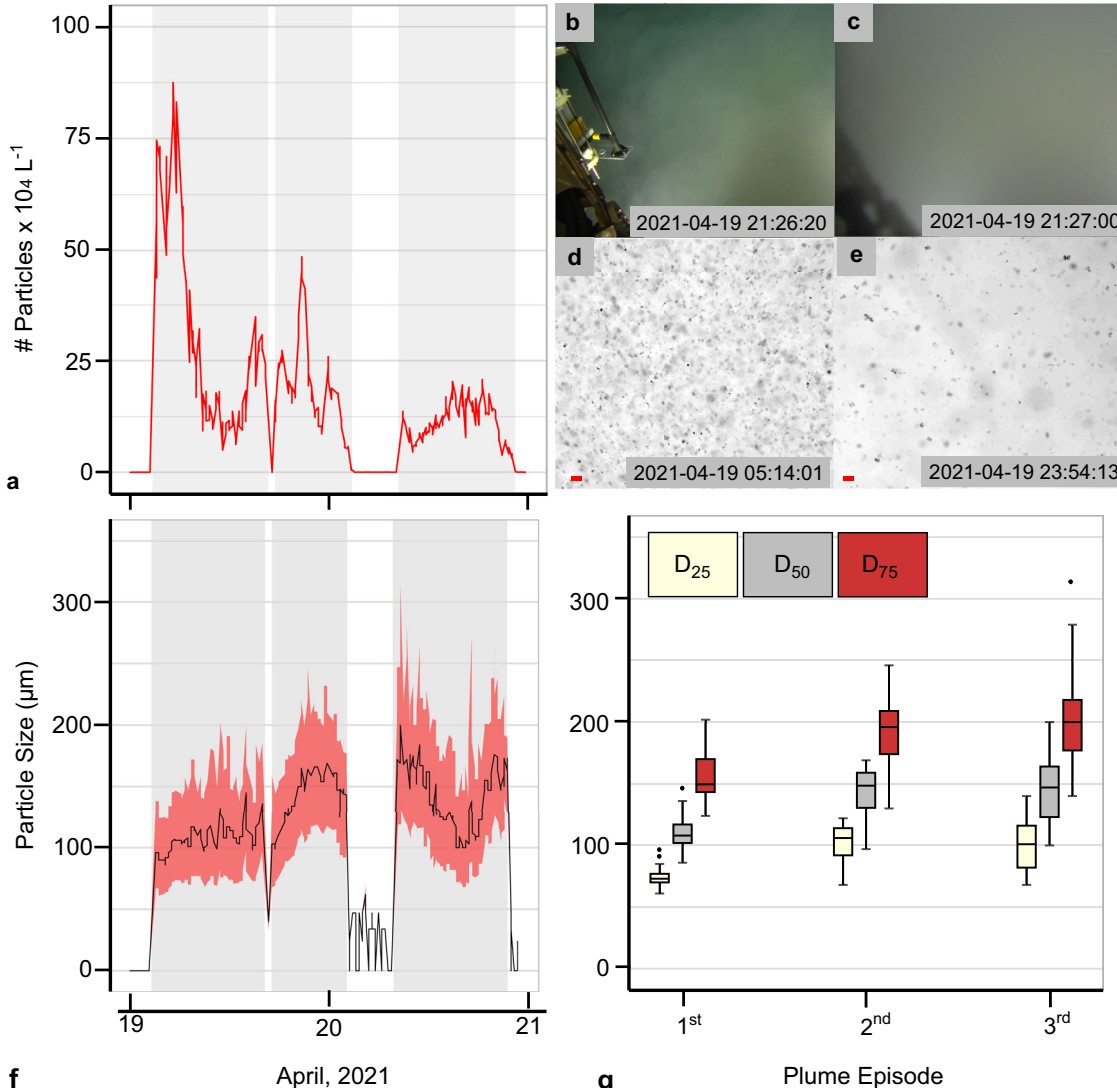

**Fig. 8 | Number of suspended particles and particle size during the mining trial.**
**a** Particle number during the mining trial using the PartiCam data (JUB_PFM-01 platform). Three successive benthic sediment plume (plume) episodes (grey areas) corresponding to the three mining strips were recorded. **b** Video frame grab acquired at 4.7 m altitude above the bulk of the plume. **c** Video frame grab acquired at 4.3 m altitude but now within the plume at the same location (JUB_PFM-01 platform) 40 s later. The reduced visibility becomes noticeable, with the sensor (Microprofiler)[2] hanging from the arm of the remotely operated vehicle (ROV) at about 1 m from the camera not being visible in (**c**). **d**, **e** In situ PartiCam images during the first day of the mining trial show a decrease in the number of particles

and an increase in particle size with time. The red scale bar is 0.27 mm. The minimum particle size that could be detected with that optical setup was 0.0153 mm with a pixel resolution of 0.0051 mm. **f** Particle size time-series distribution during the mining trial. The black line corresponds to the median particle size ($D_{50}$), and the pink area corresponds to the first ($D_{25}$) and third ($D_{75}$) particle size quantile range. **g** Boxplots of the particle sizes $D_{25}$, $D_{50}$ and $D_{75}$ for the 1st ($n = 92$), 2nd ($n = 66$) and 3rd ($n = 100$) plume episodes ($n = 258$ observations). The black centre line is the median value (50th percentile), and the boxes contain each dataset's 25th to 75th percentiles. The black whiskers mark the 5th and 95th percentiles, and values beyond these bounds are outliers (black dots).

meaningful and tailored adaptive environmental monitoring during future mining activities. Long-ranging AUVs (endurance of 90 h or more) with integrated multi-sensor instrumentation, such as the one used in this study, support limited near-real-time data communication, e.g., turbidity and position data, allowing an adaptive change of the AUV mission path based on the interpretation of the incoming data. Acoustic communication, small-cabled sensor networks, or repeated upload of AUV data at central data nodes can already be utilised today. Flexible decision-making strategies that are refined based on the incoming environmental data are already applied in the offshore dredging industry, and similar schemes have been proposed for deep-sea mining[66]. Towards automated plume monitoring, existing studies on AUV deep-sea hydrothermal plume tracking[67] could be adjusted for deep-sea nodule mining.

Secondly, the most difficult challenge is to define quantitative tolerance threshold levels of representative deep-sea biota to plume concentrations and thickness of sediment redeposition as a measure to protect the wide range of benthic life inhabiting the seafloor areas targeted by mining. What tolerance limits do deep-sea fauna have for suspended and redeposited sediment of certain particle size distribution and chemical composition, and for what time of exposure before distress or mortality occurs[68,69]? Relating impacts on benthic life observed after the trial to different levels of exposure to suspended and redeposited sediment, as derived from our study, is a first step in that direction but requires time.

This in situ study is part of a worldwide effort to increase our understanding of deep-sea mining and the requirements that future monitoring technologies should meet at a time that the ISA is

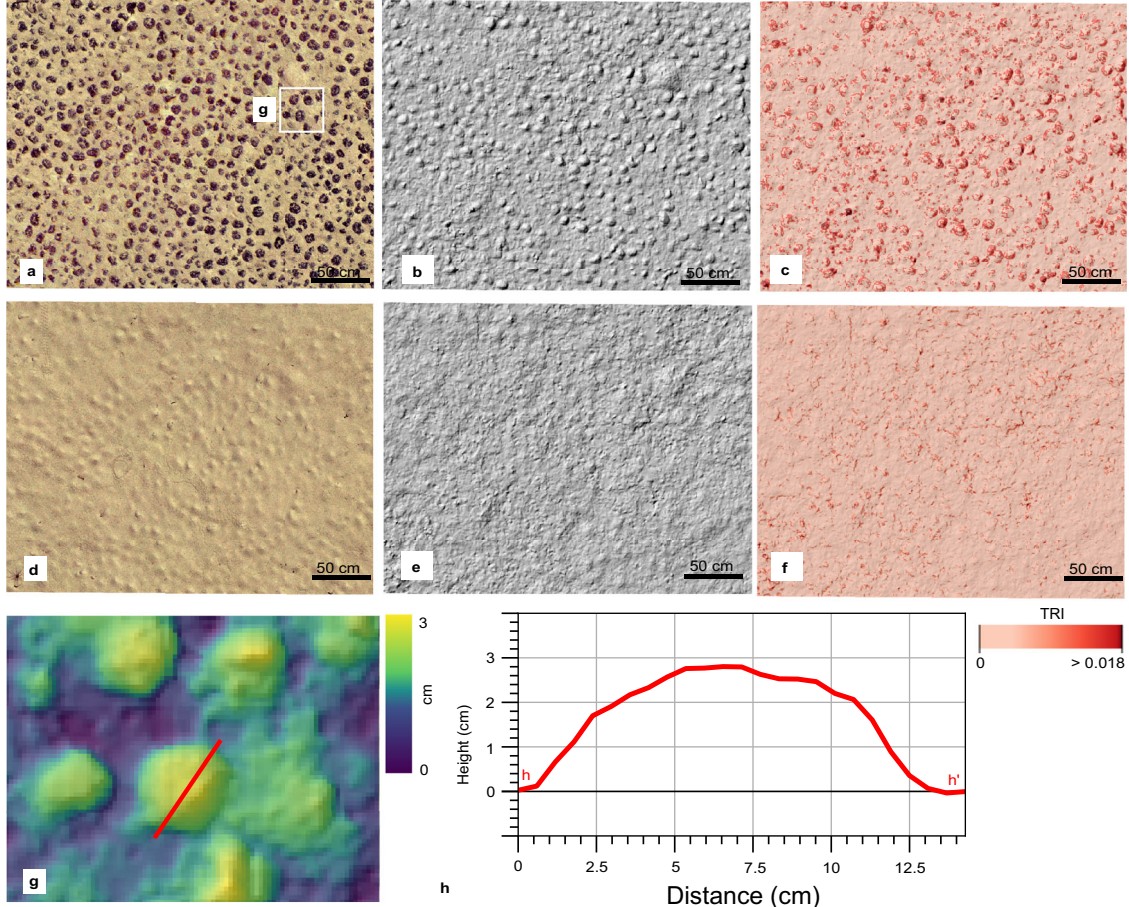

**Fig. 9 | Sediment redeposition within the impact site. a** Seafloor orthophoto-mosaic before the mining trial within the impact site. The exact location is shown in Fig. 10. **b** Hill shade of the same seafloor part based on the image-derived digital elevation model (DEM). **c** Terrain Ruggedness Index (TRI) of the image-derived DEM (see Methods). **d–f** Orthophoto-mosaic (**d**), hill shade (**e**), and TRI (**f**) of the same seafloor part after the mining trial. The sediment redeposition has covered the nodules entirely (**d**) and smoothed out the seafloor cm-scale relief (**e–f**). **g** Image-derived DEM of the seafloor before the mining trial showing large nodules and a height profile over a large nodule (red line). **h** The height profile shows that the exposed part of the nodule reaches almost 3 cm above the seafloor before the sediment redeposition and is covered with redeposited sediment after the mining trial (**d**).

draughting regulations for the exploitation of deep-sea minerals[6], and several nodule collectors are under development[18].

## Online video content

A 4D visualisation of SPM concentration (mg L$^{-1}$) evolution is accessible here: https://cloud.geomar.de/s/6cZC4Mg8ZPFQiFG

A timelapse video showing the first two plume episodes is accessible here: https://cloud.geomar.de/s/BJkHskFmG47onei

Part of the ROV video footage during the mining trial is accessible here: https://www.youtube.com/watch?v=wVGmmfqkItA&ab_channel=GeoChannelBGRLBEG

https://www.youtube.com/watch?v=EQ0J5hd2EPY&ab_channel=GeoChannelBGRLBEG

A video on resuspension and bedload transport of recently settled plume sediments at critical shear stress of 0.04 N m$^{-2}$ is accessible here: https://cloud.geomar.de/s/6qR8Df95n25fB6d

## Methods

### AUV MBES & SSS data

A HUGIN 6000 AUV (© Kongsberg SA) was used to acquire MBES, SSS, OBS data and seafloor images. Detailed mission objectives, survey routes and statistics are provided in the expedition report[2]. Multibeam bathymetric and backscatter data from the seafloor and the water column were acquired using the Kongsberg EM2040 at 400 kHz. The

description of the acquisition system settings (e.g., opening angle, beam spacing pattern, pulse mode, AUV altitude) is provided in the expedition report[2]. The raw data post-processing was done with QPS Qimera v1.7 and included full-depth Sound Velocity Profile (SVP) ray tracing correction, roll bias correction, turn removal, filtering of erroneous soundings, and tide correction based on the pressure sensors (Digiquartz®) of two SBE16 CTDs at fixed seafloor locations[2]. The navigation data was post-processed onboard using the Kongsberg SA proprietary NavLab software. The absence of an absolute navigational shift was confirmed using the processed ship-based MBES data[70] as a reference layer. The final bathymetric grid was projected on the Universal Transverse Mercator (UTM) zone 10 N and exported as GeoTIFF with a 2 m × 2 m cell size. The MBES WCI data were processed with FMMidwater v7 (built-in workflow) and exported as an ASCII dataset. In parallel to MBES mapping, the EdgeTech 2205 SSS running at 230 kHz acquired SSS backscatter data[2]. Data post-processing was done with the CT SonarWiz v7.09 software and included bottom tracking (built-in threshold detection method), gain correction (built-in Empirical Gain Normalisation method), nadir correction, and turn removal. The high overlap between the neighbouring lines resulted in the absence of nadir gaps, while the final mosaicking was done using the average method for blending. A 22 m navigation drift to the west was measured between the SSS and MBES grids, as they have been recorded on different dives. The SSS grid was corrected accordingly using the ESRI

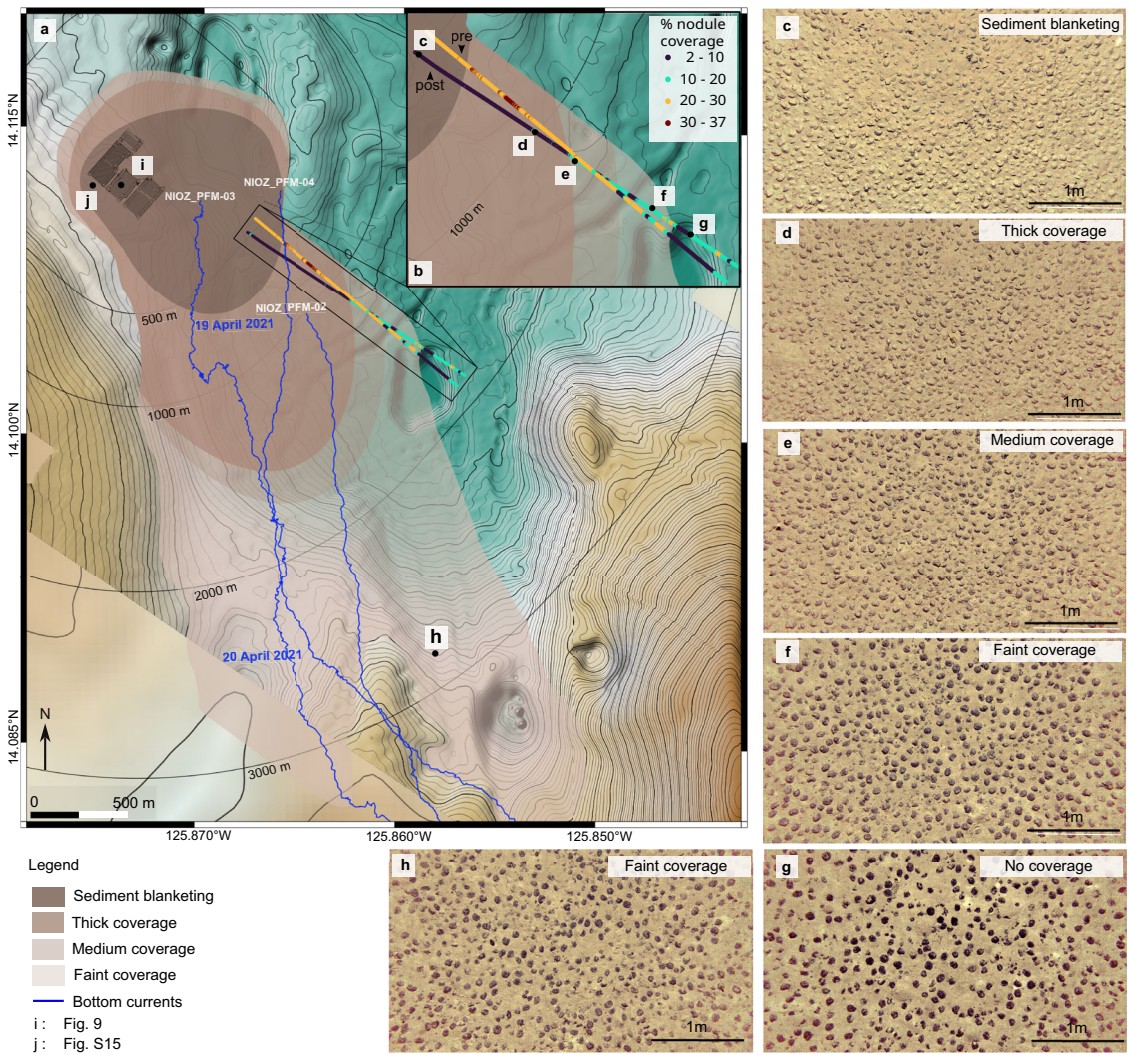

**Fig. 10 | Sediment redeposition map for the entire study area. a** The extent of the redeposited sediment based on AUV and ROV footage. A classification scheme of sediment blanketing followed by thick, medium, and faint coverage was used (see "Methods"). The bottom current trajectory plots from three acoustic doppler profilers (ADCPs) are shown in blue. The timestamp is also given for the trajectory plot of NIOZ_PFM-03. **b** The degree of sediment redeposition expressed as % of seafloor nodule coverage (see "Methods") is shown before (pre) and after (post) the passage of the benthic sediment plume. **c–g** Autonomous underwater vehicle (AUV) images showing reductions of sediment coverage at successive distances in the SE direction. An extensive version of (**c–g**) is given in Supplementary Fig. S17. **h** AUV image of a faint sediment coverage, 2.5 km S-SE from the impact site. **i** Location of Fig. 9. **j** Location of Supplementary Fig. S15.

ArcMap Georeferencing Toolbox. The final backscatter grid was projected in the Universal Transverse Mercator zone 10 N (UTM 10 N) and exported as GeoTIFF files with 0.20 m × 0.20 m cell size.

## AUV Image data

An AUV HD CathX Colour Still Camera (M12 A1000) with a resolution of 12.5MP (4096 × 3072) was used to acquire seafloor images. The acquired images were colour corrected/normalised using the DeepSea Monitoring group (GEOMAR Helmholtz Centre for Ocean Research Kiel) in-house software Image Normalisation[71], which removes lighting artefacts and illumination cones created in underwater images while aiming to restore the original seafloor colour. The colour-corrected images from the impact site were used to reconstruct the seafloor orthophoto-mosaics and DEM. The Agisoft Metashape Pro v8.3 software was used following the built-in workflow (Image alignment, Dense Point Cloud creation, Dense Point Cloud Confidence filtering, DEM, and orthophoto mosaic reconstruction). As the AUV camera was provided without underwater calibration parameters, the Agisoft Metashape Pro built-in camera calibration workflow was used

to estimate the camera orientation parameters (e.g., the principal point coordinates). Due to the large number of images (≈ 500, 000), the photomosaic area was divided into chunks of 20,000 images to make computation feasible. The chunk alignment was carried out based on shared features inside the overlapping parts. The absolute seafloor position was corrected where necessary based on the seafloor and mining features derived from the MBES and SSS data using the ESRI ArcMap Georeferencing Toolbox. The final orthophoto-mosaics and DEMs were exported in 2 mm × 2 mm and 5 mm × 5 mm cell sizes, respectively. The TRI was calculated using the Riley formula[72] as applied in the free and open-source software QGIS v3.24, which was also used to create all maps presented here.

The percentage of nodule seafloor coverage was quantified using the saltation GmbH & Co. KG proprietary software Mangan Analyser, fine-tuned for this explicit dataset. The software is based on a Hierarchically Growing Hyperbolic Self-Organising Map (H²SOM) approach to segment and detects the % of seafloor nodule coverage per square metre[73]. The algorithm has been used in the past to quantify nodules' spatial distribution and support resource estimation studies[74–76]. The

same methodology has been applied to quantify the spatial extent of sediment redeposition during an epibenthic sledge disturbance experiment in the CCZ[13]. The difference in detectable % seafloor nodule coverage between the seafloor images before and after the plume passed and (partially or totally) covered up the nodules shows the spatial extent and degree of sediment redeposition. Such an approach works well when the redeposited sediment completely covers the polymetallic nodules (i.e., 0 % detection) but not when there is a faint sediment cover (powder) on top of the nodules before the additional sediment deposition[66,75]. The latter is not the case in our study, as shown in seafloor images acquired before the trial (e.g., Supplementary Fig. S16). The images were obtained along two AUV transects in the SE direction ≈ 29 h before and ≈ 72 h after the mining trial. The two transects have a similar length of ≈ 1400 m and a maximum distance of ≈ 85 m between them. Only the images acquired in flying altitudes 5–7 m were analysed as they have better quality characteristics (e.g., brightness, contrast). Another limitation is that taking images from the exact same area of seafloor with an AUV is difficult due to slight (cm- to m-scale) navigation uncertainties. In our case, we use images from survey lines that cross each other at a slight angle due to operational restrictions during the AUV dive: two ROVs were used at the same time to recover equipment on the seafloor, and the AUV needed to keep a safety distance, prohibiting flying the same survey lines as before the plume settled. In total, 2678 and 2710 images were analysed before and after the mining trial. The results were exported as an ESRI point shapefile and projected in UTM 10 N. Parallel to detectable % seafloor nodule coverage, qualitative criteria were used to distinguish the degree of sediment redeposition. <u>Blanketing</u>: Nodules are entirely covered by redeposited sediments. The open spaces between neighbouring nodules are filled by redeposited sediment. The nodule shape is not distinguishable, and only the nodules' top part, protruding slightly from the seafloor, is still visible. The AUV seafloor photos are yellowish, and the nodules' dark (black) colour is not visible anymore. <u>Thick</u>: Nodules are entirely covered by redeposited sediments, but their shape is distinguishable. The nodules' dark colour is still not visible. <u>Medium</u>: Nodules are partly covered by redeposited sediments. The nodules' dark colour is still discernable. <u>Thin</u>: Nodules are partly covered by redeposited sediments. The brighter sediment slightly shades their dark colour.

## OBS data

Four different types of OBS were mounted on the stationary seafloor platforms[2]: (i) stand-alone JFE Advantech Infinity Series ATUD-USB (JFE), (ii) stand-alone AQUAlogger® 310TY (Aqualogger), (iii) WetLabs ECO FLNTU (FLNTU) coupled on SBE16 and SB19 + CTDs and (iv) Seapoint STM-S (Seapoint) coupled on SB19 + CTDs. The JFE works at 800 nm, with a 0–1000 FTU range, precision of 0.03 FTU and accuracy of ± 0.3 FTU ( ± 2 %). The AQL works at 880 nm, with a 0–10,000 FTU range, precision < 1 FTU and accuracy of < 2 %. The FLNTU works at 700 nm wavelength, with a 0–25 NTU measurement range, sensitivity of 0.013 NTU and precision of ≈ 0.0065 NTU. The Seapoint works at 880 nm, with a 0–25 NTU measurement range, sensitivity of 200 mV/ FTU (at 100x gain) and linearity of ± 2 %. The acquisition settings for all sensors are provided in the expedition report[2]. The AUV carried two OBS: (i) a fixed-mounted FLNTU coupled on a SAIV A/S SD208 6 K CTD and a stand-alone JFE attached on the AUV dorsal fin[2]. The JFE raw data had an offset of one FTU in clear bottom water compared to the calibration dive and the lab calibration. The JFE sensor window was checked and cleaned before each dive, ensuring no sediment, dirt, or damage affected the measurement. The sensor recorded 0.04 FTU in clear CCZ calibration water during the lab calibration, showing no sensor drift over time. A reasonable explanation for the offset is that something (part of the AUV fin, security rope, or a loose end of the tape) was in the sensor's field of view during the AUV dive. Based on comparisons with the FLNTU data, the offset occurred exclusively for

lower concentrations. A hand-fitted offset of 1 (0.98–1.04) FTU was corrected, resulting in a good match with the FLNTU data ($R^2 = 0.94$).

## OBS calibration

All OBS were calibrated to reference suspensions prepared from CCZ surface sediment (top 5 cm of the multicore IP21-070MUC)[2] dispersed in clear CCZ bottom water (IP21-072CTD)[2] at a temperature of 1.5 °C. The principle of calibrating sensors by reference to artificially prepared sediment suspensions has been described earlier by Guillen et al. (2000)[77] and for CCZ sediment by Haalboom et al[12]. Briefly, the suspensions were contained in a cylindrical calibration vessel made of PVC, sprayed matt black inside to reduce light reflections. A stirring rotor and submerged aquarium wave pump kept the suspension homogeneous, while baffles prevented the formation of a vortex that might create air bubbles. The lid on top had a porthole through which the sensors were inserted, shielded from incident light. For the first calibration step, the vessel was filled with 16 L of CCZ bottom water with an SPM concentration of 0.14 mg L⁻¹. Sensors were immersed one after the other in the water for 1 min while measuring at a rate of 1 Hz. For the next calibration step, half the volume of water in the vessel was replaced with an equal volume of prepared sediment suspension, producing a suspension with a concentration of 726 mg L⁻¹. In each of the ten subsequent steps, half the volume of suspension in the vessel was replaced by an equal volume of CCZ bottom water, thus producing a sequence of logarithmically decreasing concentrations. The measurements were averaged per calibration step for each sensor, plotted against the concentration of the reference suspensions and fitted with a linear function of the form $OB = c \times SPM$, where OB is optical backscatter recorded as FTU for the JFE/Aqualogger and as output voltage for the FLNTU/Seapoint sensors; SPM is the concentration in (mg L⁻¹) and c the: regression constant. For all JFE, the $R^2$ of the regression was 1.0000. For all other sensors, $R^2$ was > 0.9980. Turbidity recorded in the field was converted to SPM concertation (mg L⁻¹) by application of the respective regression functions determined for each sensor. Since the sediment plume investigated in our study was artificially produced by the mining machine stirring up fine-grained seafloor sediment into the bottom water, suspensions produced in the lab from seabed sediment and bottom water collected from the same region provide a good analogue for calibrating sensors in the lab. The data quality was assessed by a feasibility check to identify spurious values or instrumental drift. Two FLNTU sensors on GMR_PFM-47 and GMR_PFM-48 had an offset of + 0.17 NTU and + 0.175 NTU, respectively, which was subtracted. For the JFE, internal quality control was used to discard erroneous data. Data recorded during the platforms' deployment, relocation, and recovery were dismissed. The AUV FLNTU sensor, belonging to MV Island Pride, was intercalibrated as it remained onboard and could not be calibrated with in situ sediment at land. Using the relationship between SPM concentration (mg L⁻¹) and optical backscatter (FTU/NTU) established during lab calibration of the JFE sensor and the relationship between AUV JFE and AUV FLNTU data, the AUV FLNTU turbidity data could be converted to SPM concentration (mg L⁻¹). The total least squares – orthogonal regression (TLS) was used to calculate the relationship between the corrected and calibrated AUV JFE and AUV FLNTU turbidity values. TLS was preferred over the commonly applied Ordinary Least Squares (OLS) as TLS accounts for errors in independent and dependent variables. The AUV FLNTU turbidity in SPM (mg L⁻¹) was found to be $SPM_{\text{FLNTU}} = \frac{OBS_{FLNTU} - 0.02278291}{1.0415194 \cdot c}$, with c being the regression constant.

## PartiCam data

The PartiCam is composed of two separate pressure housings, one containing the camera and one containing the flash. It was equipped with a Canon EOS 760D SLR with a resolution of 24.2 megapixels and a Canon EF-S 60 mm f 2.8 macro lens. Using a macro lens allows the images to be taken closer to the lens, providing a higher optical

resolution and a more even illumination of the sample volume. A collimated light source (Yongnuo YN-468 II Speedlite strobe) was lined up and facing the camera with a ≈ 2 cm slit between the flash and camera window through which water could flow. A short flash duration of 1/200 s allowed the acquisition of images containing particles that were in focus without motion blur effects. This optical setup's minimum detectable particle size is 0.0153 mm, with a pixel resolution of 0.0051 mm. The MATLAB Image Processing toolbox[78] was used following the methodology of Iversen et al.[79], and the ParChar architecture code[80] was run to remove the image background and extract the size of the particles in each image. A rolling median ($k = 3$, with k being the integer width of the rolling window) was applied to remove a few isolated outliers from the raw data and allow better visualisation of the trend of increasing $D_{25}$, $D_{50}$, and D75 during the mining trial. The R zoo package was used[81]. The raw data before and after applying the rolling median are shown in Supplementary Fig. S14.

### ADCP data

In total, 5 Teledyne RDI Workhorse Sentinel ADCPs at 300 kHz, one at 1.2 MHz and 7 Nortek ADCPs at 2 MHz were deployed on fixed seafloor platforms. A detailed overview of the sensor specifications and acquisition settings is provided in the expedition report[2]. Data from 10 devices are presented here (see Supplementary Material). Firstly, the data recorded during platform deployment, recovery, or any relocation by the ROV were removed. Secondly, the data in bins where the acoustic backscatter was lower than 25 counts for high-frequency (2 MHz) ADCPs and 40 counts for other lower-frequency ADCP sensors were discarded to ensure the reliability of measured current speed and direction. The manufacturers provide these values as the noise floor, i.e., the lower backscatter threshold, to ensure good data. In addition to the set ensemble averaging performed internally by each device during the measurement, the data were averaged over the user-defined burst duration. The measured raw acoustic backscattered signal in counts was converted to echo strength (dB) and corrected for sound attenuation away from the source by both acoustic spreading and absorption by water[82]. Due to the naturally low concentration of suspended particles in the deep ocean water column[12,17,83], even during the plume episodes (mostly below 200 mg L$^{-1}$), the sound attenuation by the suspended particles was assumed to be negligible[84].

### Sediment transport estimation

The AUV MBES WCI data captured the plume's lower part (< 5 m) during a total time of 3.5 h (19 April 2021; from 08:40 to 12:05) on the first day of the mining trial (1st strip, Fig. 3 and Fig. 5). The recorded data showed a variable spatial extent while the plume was spreading over a maximum distance of ≈ 700 m, which was recorded continuously for the last two hours (Fig. 5). The 2 MHz ADCP at NIOZ_PFM-04 showed that the average current direction and magnitude at 2.25 m altitude when the plume was passing by, was 117° and 0.062 m s$^{-1}$ respectively. According to the OBS data at NIOZ_PFM-04 and RBNIS_PFM-01, the average concentration was 18.37 mg L$^{-1}$ at 1 m altitude. The cross-sectional area defined by the MBES WCI maximum spatial extent along the AUV flight path was nearly normal to the plume axis (defined by the average current direction) with a 22° deviation from this. It has a length of 640 m, a height of 5 m and a 95° angle between its normal vector and the N (Supplementary Fig. S18). We assumed a constant concentration, current direction and magnitude (equal to the above values) for the entire cross-sectional area of the first two metres of altitude during the two hours with the maximum recorded plume spread. The altitude of 2 m is based on previous work[2], which showed that 92 to 98% of the sediment mass remains below 2 m. The nodule collector average flow rate was also assumed constant for these two hours (12 kg s$^{-1}$)[20]. The equations used for estimating the

sediment transport are as follows:

$$\Phi = SPM_c U_{ADCP} \cos \alpha \qquad (1)$$

$$F_r = \Phi A_{cs} \qquad (2)$$

$$M = F_r \Delta t \qquad (3)$$

$\Phi$ : Particles flux[kg m$^{-2}$ s$^{-1}$]
$SPM_c$ : Suspended particulate matter[kg m$^{-3}$]
$U_{ADCP}$ : Current magnitude given by ADCP measurements[ms$^{-1}$]
$\alpha$ : Angle in degrees between the gravity current direction (ADCP measurement) and the unit vector normal to cross − sectional area
$F_r$ : Particles flow rate across cross − sectional area[kg s$^{-1}$]
$A_{cs}$ : Cross − sectional area[m$^2$]
$M$ : Sediment mass[Kg]
$\Delta t$ : Time window[sec]

The same equations were used to calculate the sediment mass at 5 m altitude along the same AUV flight path 500 m from the impact site. The two AUV OBS data sets revealed a similar lateral extent as the MBES WCI (slightly extended to SE direction), corroborating the hydroacoustic results (Fig. 3). The concentration spatial distribution along the AUV flight path is similar to the MBES WCI data, with higher concentration in the eastern part (Fig. 3). The average concentration along this pathway is 3.15 mg L$^{-1}$. When the plume was passing by, the average current direction and magnitude at 5.25 m altitude were 133° SE and 0.06 mm s$^{-1}$, respectively. As with the sediment mass calculation above, we assumed constant concentration, current direction and magnitude for the entire cross-sectional area between 5 and 6 m altitude during the two hours of plume recording.

### Reporting summary

Further information on research design is available in the Nature Portfolio Reporting Summary linked to this article.

## Data availability

All data associated with the results of this study are presented in the paper, Supplementary Material and online video content. The data will be published on PANGEA® Data Publisher for Earth & Environmental Science (https://www.pangaea.de/) and can be obtained from the corresponding author upon personal request.

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

## Acknowledgements

We gratefully acknowledge the captain and crew of MV Island Pride for their assistance during the MANGAN 2021 expedition. We thank the Ocean Infinity ROV and AUV teams for extensive dives. We are grateful to the German Federal Institute for Geosciences and Natural Resources (Bundesanstalt für Geowissenschaften und Rohstoffe - BGR) for inviting MiningImpact 2 scientists to join their research cruise on MV Island Pride and for chartering both ROVs. We are thankful to the Global Sea Mineral Resources NV (GSR) / DEME Group, who chartered the AUV onboard MV *Island Pride*. We express our gratitude to Francois Charlet (GSR/ DEME Group) for his support during the MANGAN 2021 expedition, particularly for the communication between the GSR team onboard MV Normand Energy and our team onboard MV Island Pride. Special thanks go to the Oceanic Machine Vision Group at GEOMAR for their valuable recommendations on the AUV photomosaic surveys. We thank the GEOMAR Library team for its support in gathering the bibliography and Anastasios Poulos-Sidiropoulos for his assistance with the MBES and SSS data post-processing. This research was carried out in the European collaborative project MiningImpact 2 and received national funding through the Joint Programming Initiative Healthy and Productive Seas and Oceans (JPI Oceans): German Ministry of Research grant no. 03F0812A-H; Dutch Research Council grant no. 856.18.002. Additional funds for representing the data within the 4D Digital Earth Viewer came through the Helmholtz Project "Digital Earth" grant ZT-0025. M.D. acknowledges funding through the Plumefloc project; Dutch Research Council grant no. TWM.BL.019.004. This is publication # 68 of the DeepSea Monitoring Group at GEOMAR Helmholtz Centre for Ocean Research Kiel.

## Author contributions

All authors collaboratively contributed to the conception and design of the study. H.S., M.H., A.V., M.B., B.G., L.T. and J.G. offered scientific equipment. I.Z.G., H.S., J.M., K.H., M.H. and A.V. acquired the data. I.Z.G., H.S., J.M., K.H., M.D., B.G., L.T., M.B. and M.V.A. contributed to the data post-processing. I.Z.G., M.D. and M.V.A. collaboratively contributed to the data visualisation. J.M., K.H., L.T. and A.V. collaboratively contributed to the creation of video data. I.Z.G., H.S., M.V.A. and K.P. collaboratively contributed to the sediment transport estimation. M.H. acted as EU MiningImpact 2 project leader. L.T. and J.G. acted as EU work-package leaders for this monitoring mission, and L.T. was significantly involved in finalising the manuscript. I.Z.G. compiled the results and wrote the manuscript. I.Z.G., L.T., H.S. and J.G. revised the manuscript. All co-authors commented on the manuscript and provided input to its final version.

## Funding

## Competing interests

The authors declare no competing interests.
