## [Peer Review File · Nature Communications]

Monitoring benthic plumes, sediment redeposition and seafloor imprints caused by deep-sea polymetallic nodule mining

Corresponding Author: Mr Iason Gazis

Version 0:

Reviewer comments:

Reviewer #1

(Remarks to the Author)

This manuscript presents a novel in-situ monitoring approach for studying the imprints and plume generated during deep-sea polymetallic nodules mining. The research is of great interest to the readers of Nature Communications. The manuscript provides a detailed description of the experimental steps and pathways followed during deep-sea mining experiments. By employing in-situ monitoring techniques using lander, ROV, and AUV, the authors were able to obtain valuable information regarding the rutting marks caused by the mining car, as well as the migration and distribution patterns of the mining plume concentration. However, there are some areas of confusion in the manuscript. If the authors can address and clarify these issues appropriately, and further refine the structure of the manuscript, I would recommend accepting it for publication.

My concerns about this manuscript mainly include:

1. The title of the study refers to the imprint and plume observed in a mining experiment. However, the article only discusses the rutting imprint as a form of marking, without considering other potential imprints. Additionally, it is unclear whether the thick redeposition of the plume can be considered as an imprint. It is recommended to provide a clear definition of these terms in the article. Furthermore, the discussion section lacks a discussion on imprinting, which should be included.

2. The abstract of this paper includes numerical values that are specific to certain working conditions. This may lead to misunderstandings among readers. It is suggested to provide a more general conclusion. For instance, the abstract mentions that the plume migrates farther at a slope of 3° compared to a flat area. However, this observation is only valid under specific flow field conditions. Plumes can still be observed at distances of 4.5 km, 500 m, and 50 m. It would be beneficial to include information on the change rate of plume concentration at these distances relative to the initial concentration value. Such data would provide a more comprehensive understanding and should be carefully considered.

3. This manuscript exhibits characteristics more akin to a report rather than a scientific paper. The objective of the experiment is not clearly stated, and the problem being addressed is not adequately reflected in the manuscript. Such deficiencies are clearly unreasonable for a scientific paper.

4. The length of the manuscript is excessive. Certain sections within the proposed methodology can be omitted, while additional content can be appropriately included in the supplementary materials. For instance, the description of the full water depth velocity profiler in Line 656 does not appear to be utilized in this paper and is recommended for deletion. Similarly, the content spanning Line 765-782 would be better suited for inclusion in the supplemental materials.

5. In Line 164-167, the presence of gravity currents suggests the potential resuspension of seafloor sediments, which necessitates further interpretation. It may be possible to calculate the critical shear stress associated with this phenomenon.

6. In Line 317-318, the utilization of the coverage of polymetallic nodules following sediment cover as a means to characterize the extent of plume cover is a commendable approach. However, this method becomes impractical when the sediment completely covers the polymetallic nodules. Additionally, accurately determining and calculating the cover area becomes challenging when the sediment cover area is small.

7. In the discussion from line 374 to 382, it was mentioned that flocculation occurring during plume settlement leads to an increase in the median particle size from 12 μm to 470 μm . It is suggested that the sedimentation rate of particles after flocculation should be included for a more comprehensive analysis. Additionally, it is recommended to incorporate the basic physical parameters of surface sediments in the study area, such as density, water content, pore ratio, and consolidation coefficient, into the paper.

8. From line 383 to 387, it is stated that the underflow becomes the driving force of the plume motion after a distance of 500m. However, it is unclear what the driving force is before reaching this distance. If the underflow is not the driving force, the reason behind the plume moving in the ES direction should be further explored. The discussion in this section lacks clarity in conveying the author's intended message.

9. Line 716-751. The manuscript describes the preparation of suspended particle liquids with varying concentrations using soil samples collected from the surface of the study area. These liquids were then tested for turbidity in the laboratory to establish the relationship between NTU and mg/L. Typically, to ensure the reliability of such tests, this relationship is established based on the results obtained from seawater sampling at different CTD levels. It is important to understand the differences between the results obtained in this study and those from in-situ testing. Furthermore, the experimental setup in this study bears similarities to a previous experiment conducted by Munoz-Royo et al. (2022). The attachment in this paper provides a detailed description of establishing the relationship between NTU and mg/L, which is recommended as a reference.

Muñoz-Royo, C., Ouillon, R., Mousadik, S.E., Alford, M.H., Peacock, T., 2022. An in Situ Study of Abyssal Turbidity-Current Sediment Plumes Generated by a Deep Seabed Polymetallic Nodule Mining Preprototype Collector Vehicle. *Science Advances*.

10. There are some minor errors in formatting and spelling in the manuscript, such as:

Line 192 "1800m" should be "1800 m".

Line 687 "absoluteseafloor" should be "absolute seafloor".

Line 790 62.33mm s-1, should be "62.33 mm s-1".

Fig. 1. Only what is shown in figures b and c is marked. The section shown in Fig c is the a1-a2 line in Figure d, which is difficult for readers to read in order.

Extended Data Fig. 2. The horizontal coordinate in the figure is not marked, although I know it is the date, but it does not meet the specification of drawing.

Reviewer #2

(Remarks to the Author)

The manuscript describes plume monitoring during a pilot test of deep sea mining equipment. This study is unique in its kind and provides a significant contribution to quantify the environmental impact that might be resulting from seabed mining activities.

It is highly appreciated that the monitoring campaign is conducted independently, i.e. through government funding, supporting transparency and openness.

The authors provide extensive explanation and supporting evidence of how they have conducted their research. The manuscript provides good insights that are complementary to the near field measurements conducted by MIT. Together with the various sorts of data collected, the authors have managed to provide a clear picture of what most likely has happened during these tests. In most cases, sufficient explanation has been provided to prove or to philosophize what mechanisms or probable causes determining the main outcomes of the pilot vehicle test.

I would recommend that this manuscript will be published, after the feedback has been addressed. This research is unique and of very high relevance towards responsible use of the oceans and exploitation of the seabed.

General remarks:

1. The study only mentions the turbid flow resulting from the discharge of the excess of water and sediment flowing through the collector vehicle. Although it is the main source of turbidity, the sediment kicked up by the caterpillar tracks and the sediment spilled around the collector device can still be a significant amount. Please reflect on this.

2. Line 104/105: are all three strips driven from SW to NE? If so, would be good to reflect on this in the discussion, as currents moving downslope of new lanes might catch up existing currents. Catching up of turbidity currents has not been thoroughly studied and thus this might be an interesting option to be considered for mitigation of impact (potentially preventing the ripples).

3. Fig 2b3: Why is it that the depth along the centerline is taken as 0? If the profile is taken slightly longer, then the 'virgin' bed outside the lane could be used as a reference of 0. That would also give a more fair interpretation of the amount of deformation underneath the tracks (compaction + scooping), as well as erosion depth caused by the collector device.

4. Lines 166-167: Is there any data collected that might provide insight on the propagation speed of the current? If so, how do the current velocities relate to the erosion velocity of the sediment? Is there any information available regarding the erosion velocity of the 'virgin' seabed and that of the freshly deposited material. This might be relevant because to identify whether mitigation might benefit from longer time intervals between neighbouring lanes.

5. Line 167-168: Based on the map in fig1, it might have been interesting to consider similar mapping from area 3 to mooring NIOZ_PFM-03, to identify whether sediment ripples would have occurred parallel to the lanes. Is there data collected?

6. Line 171-172: Would this have been an effect of a single turbidity current, or is it caused by multiple turbidity currents catching up on each other along the slope, originating from multiple lanes? Would you eventually get a larger single gravity current, or multiple?

7. Line 274: statistic of 0.03-0.47mm does not match with fig 5b. Based on what is visible in that figure, $d_{50} < 0.06$ does not occur. Furthermore, I would suggest to include data of NIOZ+PFM-03 next to this as well, to help interpret the particle number to concentration. Furthermore, the figures would probably improve if a distinction would be made between when the plume passes by and where there is no to little plume (particle count is very low, but PSD seems quite spiky in that interval). 5b shows a few peaks $>400\mu$. Is there any explanation why these spikes are observed? And the numbers on y-axis are not readable and not properly positioned.

8. lines 304-308: Suggest to include that this is a conservative calculation. As the vehicle moves along the mining strip, it disturbs the recently settled materials from previous lanes. Whether this influences the source term (amount of sediment entrained by the collector device), probably no such detailed measurements of the source term exist.

9. Lines 317-319: Although the qualitative trend of blanketing is clear, it would be good to explain how this is defined. What metric is used to assess the severity of blanketing? And what criteria are used to distinguish between faint, medium, thick and blanketing?

10. Discussion: Suggest to add a statement in the discussion that the observed suspended material is more than just that originating from the discharge of the mining vehicle. The caterpillar tracks bring sediment in suspension and around the collector heads not all suspended sediment will be entrained into the system. These sources of suspended sediment behave differently compared to the discharge. reference 2, showed clear indications that the turbidity originating from driving only is still quite significant.

11. Line 388-390: your observation shows a higher concentration than that is reported by the MIT study. Could there be a plausible explanation?

12. Research outcomes: Fully agree on the two main directions. However, regarding point 1, there is a significant difference between the work that has been done in this study, scientific validation and scientific measuring of the plume. For industrial applications, such a monitoring strategy is not fit for purpose. Together with the thresholds discussion, monitoring strategy will have to be adapted.

It is likely that thresholds of e.g. max SPM at distance x will be used to manage a mining activity (environmental pressure or direct measurement). However, in the light of uncertainty of the thresholds and cumulative impacts, it is essential that technologies will be developed to enable (more) rapid assessment of the status of the habitat (environmental impact, typically indirect measurement)

Detailed remarks:

1. lines 71-75: A few questions, 1) is there quantitative info available regarding the quantity of water flow, and thus sediment concentration of the discharged mixture? 2) What is the vehicle velocity? The mixture is discharged from a moving source and thus concentration and vehicle velocity can be of influence on how dense the initial gravity current will be.

2. fig 1c/d and caption: caption states 1c top and 1c bottom, but 1c bottom is 1d

3. fig 2, b2) suggest to include where the line starts and where it ends for ease of interpreting b3. Furthermore, probably it is safe to assume that the bed imprint is symmetric along the centerline of Patania II. Then I'd suggest to let the line start at the middle between the tracks.

4. line 155: Moved slowly back and forth. It is not clear to me what is meant here. Can you reword? Does it mean that the front was not always at a similar velocity, possibly due to discharge conditions (direction, velocity and action of vehicle) and deviations in slope along the trajectory?

5. caption fig 4, lines 259-260: it appears as if the effect of the flotation body is explained twice, but at different positions

6. lines 291-294: Please rewrite, without reading the methods section, it is very difficult to understand how this needs to be interpreted.

7. Line 677: this is the first and only time DSM is used. Please explain

8. Line 688: add space to absoluteseafloor

9. Line 692: Please provide the name of the software

10. extended data fig 2: Add label to x-axis. It is done in all other figures except this one.

11. extended data fig 6: suggest to include larger scale, or enable stronger contrast. When printing this would not be visible

Reviewer #3

(Remarks to the Author)

Review of Deep-sea polymetallic nodules mining imprints and plume monitored in situ

Isobel Yeo, NOC

This is an interesting paper and I enjoyed reading it. The authors are presenting data on one of the most realistic mining collection trials to date, and as such the data presented here are of interest to both scientific and industry audiences. Their study is timely and useful and I look forward to seeing it published in Nature Communications. I have a few broad comments and a number of smaller detailed ones, none of which I think should present the authors with too many problems to address. Also, I don't know if I am allowed to comment on this but I shall anyway, the author list very male. Obviously, this now has to reflect the people who have done the work, but I would encourage this group to explore why so few women were involved in the study. Overall though I think this is a fascinating piece of work and I expect the results to be used widely in the coming years as seafloor exploitation guidelines are developed.

General comments:

1. Deep Sea exploitation is an emotive issue, and the authors are to be commended on this work, which provides useful information for understanding the impacts of nodule mining on marine ecosystems. However, there are a few times in which I think the authors underplay work that has been done (e.g. in line 53). I think paper this would benefit from a brief synthesis of some of these excellent past studies and an identification of the major research and knowledge gaps, one of which, plume dispersion, this paper addresses. Perhaps by expanding the text in lines 55-58. There are also a few parts of the paper that appear to have an anti-mining stance. While I sympathise with the author's positions here, in my opinion it is not our role to apply our own feelings to scientific results (which in this case are extremely useful). It is up to the authors, but I would dial this back slightly. In particular:

- Lines 16/17 in the abstract – maybe rephrase to, “It is likely that Deep-sea mining of polymetallic nodules may start without finalised mining regulations.” Or something equally unequivocally true
- Line 52 -I wouldn't say “dire need”, I'd specifically state the research gaps that exist and need to be addressed
- Lines 473 – 475 – I don't think you can say “race to the bottom”

2. There is no reference in the main text to baseline values although I think these are alluded to at various points. Was this data collected? It is briefly mentioned that these measurements were in the hours before and after the experiment. If this is the case, I think they may be missing normal variability, and this limitation should be stated clearly.

3. I think there are two separate things that are important for monitoring impacts, the actual measurements of the physical manifestations of mining activity, which is addressed by this study and which will not vary too much between different sites, and then the threshold values for species that make up ecosystems, which will be essential for understanding the actual impact activity has. These will vary between sites. This is mentioned towards the end of the discussion, but I think it might be worth clearly separating these two out and describing this in the introduction.

4. There is no sediment characterisation data presented here other than measurement made with the PartiCam. I don't know if sediment traps were deployed, it seems likely that they would have been, although they are not mentioned in the partner study either? Sediment characteristics are important for understanding the ability of say, certain frequencies of ADCP, to detect them, as the authors say. If there are no sediment characterisation measurements of the actual plume material then I think this needs to be made clearer, because while they have calibrated their OBS the material used in this calibration is probably similar to but not identical to that kicked up in the collector experiment (I don't think they give the location of the sample used for calibration relative to this study but I may have missed it) and therefore their concentration values may not be perfect if it has a different particle size/shape distribution. If, on the other hand they have samples of the actual plume material this data should be included.

5. There are a few places where I feel like the authors could have gone further with the data they have – particularly as they are aiming at such a high-impact journal. The data here is mostly observational, and very useful, but I think more could be done. For example, I don't think they have attempted ADCP inversion or calibration, which would be of interest given they are using very high-frequency ADCPs (and I congratulate them on deploying the correct equipment to measure a fine particle plume!). ADCP data is likely to play a big role in monitoring seafloor disturbance going forward and it would be interesting to see how successfully this data can be used to replicate their other results.

In line 446 the authors refer to adaptive spatio-temporal modelling, which I agree is a challenge. I wondered why the authors have not attempted to model the plume they measured. They reference Spearman et al. (2018) which presents a model output for a smaller plume experiment, and I think a similar simulation would be useful here. They could compare their modelled plume with the sediment redistribution and it would also allow the authors to vary conditions to predict plume response, addressing the comment they make in lines 406-407 about extrapolation of these results.

6. The figures are very data rich but, for this journal, I think need some improvement. I've given more detailed notes below.

Specific comments:

Line 19 – lose “scientifically”, independent -> independently

Line 22- Has this been observed for other gravity currents? Obviously not for the abstract but interesting to know if this is the first observation of this.

Line 23 – Further than -> Beyond

Line 23/24 – A bit confused here – are the sediments beyond 500m not part of a gravity current?

Line 31 – I'm not sure this has been shown really – the sediment redeposition is qualitative. I would reword this sentence.

Line 41 – reference needed here, probably HALBACH P. and OZKARA M. (1979) Morphological and geochemical classification of deep-sea ferromanganese nodules and its genetical interpretation. In Proc. Int. Colloq. C.N.R.S. No. 289, Sur la Genese des Nodules de Manganese (ed. C. LALOU), pp. 77-88. And/or HALBACH P., SCHERHAG C., HEBISCH U. and MARCHIG V. (1981) Geochemical and mineralogical control of different genetic types of deepsea nodules from the Pacific Ocean. Mineral. Deposita 16, 59-84.

Line 52 – lose “dire”

Lines 90-93 – How were the locations for the monitoring selected?

Line 97 – If these are the only background measurements, they miss quite a lot of possible natural variability

98 – Is this a name or should there be a space in MiningImpact?

Line 103 – either here or when it's first introduced it would be useful to give a sentence or two of context – how does this collector and the operation carried out compare to the scale of an actual possible mining operation- this currently only appears in the discussion.

Line 103 – 50 m and 4-m the “-“ seems to be used variably

Line 108 – Needs a reference for this – what have other studies shown about topographic control on gravity currents?

141 – Delete ‘ before Detail

152 – I think you need to say “measured background level” if there are no long-term measurements

Line 165 – At what distance were these observed?

Line 170 – I'd also state that this is the downslope direction in this area.

Line 202 (and throughout) – sometimes en-dashes with spaces are used to separate numbers and other times not, this should be consistent

Line 230 – these videos are great!

Line 269 – what is the sensitivity of the PartiCam for detecting fine material?

Line 274 – How does this compare with e.g. Spearman et al. 2018?

Line 286 – Could this also be quantified with image-derived DEMs? There will be complicated drape over that terrain so the actual average thicknesses could be determined?

Line 298 – What is this computer-based analysis? I'd add the name even though it's in the methods.

Line 322 - I think somewhere in the discussion it would be useful to clarify if this density current behaves in a similar way to others measured on the seafloor. Maybe in the section where the authors discuss concentration decay.

Line 361 – I think particle size is important here too?

Line 368 – or higher? Or the integration of ADCPs?

Line 373: Used OBS rather than turbidity earlier

Line 406 – Like that provided by modelling?

Line 409 – In this case, but in future experiments those could be measured.

Line 413 – Maybe reference Gillard, B. et al. Physical and hydrodynamic properties of deep sea mining-generated, abyssal sediment plumes in the Clarion Clipperton Fracture Zone (eastern-central Pacific), Elem. Sci. Anth. 7, 5. <https://doi.org/10.1525/elementa.343>. Here?

Line 424 – will the resuspension of sediment draped by other tracks potentially increase the sediment discharge rate or do the authors think this effect will be insignificant?

Line 440 – I agree and I think you could do even more with this incredible data!

Line 456 – Not sure if the switch to turbidity in this section is deliberate but it would be an OBS sensor

Line 473 – Lose “dire”

Line 475 – “Race to the bottom” needs rewording

Line 508 – Reference format

Line 589 – Reference format

Line 624 – Reference format

Line 719 – Say how far away from the study region this is

Line 761 – Aha! Good this is known, I'd add this in response to my earlier comment.

Line 780 – considered -> assumed

Figure comments:

Figure 1: (a) looks like a screen grab. Resolution needs improving and the text is hard to make out. I would make this larger and higher resolution, improve the enlarged map panel to include lat/lons and a proper scale (the numbers mean nothing to most people); (c) also looks a bit pixelated but maybe that is my screen. I also question the usefulness of a single profile that is necessarily vertically exaggerated. It might be more useful to show multiple profiles or none at all – the need to compress the horizontal scale makes the slopes look far more extreme than they are; (d) The NW-SE artefacts are not all >7 degrees as stated in the caption, some appear to be 5-7. Also, I think these are maps made in QGIS – as this is an open-source software it should be referenced.

Figure 2: I think all these images are amazing, but it would be useful to have lat/lons on them. The rectangle on (a) drawn for the image in (b) is a different shape to the image shown. The scales for b2 and c2 need more explanation - these initially look like difference maps but I don't think they are. It is unclear what “top of the redeposited sediment” means in this context – I think they mean the redeposited piles of nodules? This needs to be clearer. Was there imagery collected before the survey/in undisturbed regions? This would be a GREAT comparison to show the actual impact on the seafloor. (b3) one of the grid lines doesn't reach the top of the graph, and there should be a a/a' on b2 to show the direction of the profile.

Figure 3: (b1, b2) what is the approximate field of view here? (c, d) How long after the survey were these images taken? (f) Can't really read the text on the rose diagram; (g) Resolution on this is bad again although this could be the review copy?

Figure 4: Font sizes too small. (a) this is a slightly strange way of displaying ADCP data – it could do with a bit more explanation for what is actually shown, is it clipped? I am not an ADCP expert so other reviewers may have more comments on this. I would expand the timescale on (c) so we can make out the details, especially given there is a period of no data shown.

Figure 5: Why does the scale stop at 0 on (a)? What is the sensitivity of the PartiCam? From this it looks like maybe 20 μm ? I see that this is in the methods but I'd put it here too.

Figure 6: (a) Could you timestamp the bottom current track? Also, is it not possible to be more quantitative in your measurements of sediment cover using image-derived DEMs like you did for the tracks? In (b) the nodule coverage dots could be enlarged so they're easier to make out. (b1-5) the caption says this is shown pre-and post but these images are all post? I think this refers to the lines in (b) but images would be nice. Also are these labelled the wrong way around in (b)? The line labelled “pre” appears to have thicker coverage than post? But maybe I am misreading them because they're so thin? I'd also make the colours used for 2-10 and 30-37% more different.

ED3: can some idea of scale be given? I know this is hard in 3D.

ED7: Give location

ED8: What is the field of view?

ED9: Scale?

Version 1:

Reviewer comments:

Reviewer #1

(Remarks to the Author)

The authors have effectively addressed all the concerns raised in the initial review. This paper provides a comprehensive description of the in-situ monitoring of deep-sea mining plumes, which will significantly contribute to the research on the environmental impact of deep-sea mining. I recommend accepting this paper and eagerly anticipate its publication for the benefit of our readers.

Reviewer #2

(Remarks to the Author)

I have reviewed the extensive and well structured response to all reviewers. Based on the answers provided and the suggested changes to the manuscript, I have no further questions or remarks. Therefore it is my advice to accept the manuscript for publication.

I would like to thank the authors for their constructive and clear response to improve and strengthen the manuscript and enabling a constructive debate.

Reviewer #3

(Remarks to the Author)

Second review of Monitoring benthic plumes, sediment redeposition and seafloor imprints caused by deep-sea polymetallic nodule mining (I Yeo)

As I said in my last review, I think this is a fascinating study with wide relevance and importance to the communities involved in understanding and monitoring the impacts of seabed mining and I look forwards to seeing it published in Nature Communications. I think the authors have done a very comprehensive job of addressing my comments and those from other reviews and I commend them on that. I am delighted to hear that there are other papers coming out of this study and I look forward to them. I think the additionally material and supplementary figures will extremely useful to others working on connected research. I am satisfied they have responded to my comments and the figures are much improved. I do not need to see this manuscript again and I look forwards to seeing it (and their other results) published.

REVIEWER COMMENTS & AUTHOR RESPONSES

**Reviewer #1 (Remarks to the Author):**

This manuscript presents a novel in-situ monitoring approach for studying the imprints and plume
generated during deep-sea polymetallic nodules mining. The research is of great interest to the readers of
Nature Communications. The manuscript provides a detailed description of the experimental steps and
pathways followed during deep-sea mining experiments. By employing in-situ monitoring techniques using
lander, ROV, and AUV, the authors were able to obtain valuable information regarding the rutting marks
caused by the mining car, as well as the migration and distribution patterns of the mining plume
concentration. However, there are some areas of confusion in the manuscript. If the authors can address
and clarify these issues appropriately, and further refine the structure of the manuscript, I would
recommend accepting it for publication.

**Answer**

We thank Reviewer #1 for the time and effort spent revising our manuscript. We welcome the overall
positive evaluation and have addressed the comments and concerns point-by-point.

My concerns about this manuscript mainly include:

1. The title of the study refers to the imprint and plume observed in a mining experiment. However, the
article only discusses the rutting imprint as a form of marking, without considering other potential imprints.
Additionally, it is unclear whether the thick redeposition of the plume can be considered as an imprint. It
is recommended to provide a clear definition of these terms in the article. Furthermore, the discussion
section lacks a discussion on imprinting, which should be included.

**Answer**

We intended the title to depict the in-situ plume dispersion monitoring and mapping of direct seafloor
disturbance. As imprints, we consider the rutting marks from caterpillar tracks and the humps on each side
of the track due to sideways sediment extrusion, produced by the combined effect of erosion of seabed
sediment under the hydraulic collector head of the Patania II pre-prototype nodule collector vehicle and
compression under the caterpillar tracks of the vehicle respectively. These are the mining imprints to which
the title and Abstract refer. We do not consider the sediment redeposited on the seabed or the sediment
ripples formed by the passing of gravity current as imprints. We have revised the Abstract (lines 16–34)
and Discussion (lines 369–499) accordingly. Considering the journal's limitation, we suggest the following
alternative title: "Monitoring benthic plumes, sediment redeposition and seafloor imprints caused by deep-
sea polymetallic nodule mining".

2. The Abstract of this paper includes numerical values that are specific to certain working conditions. This
may lead to misunderstandings among readers. It is suggested to provide a more general conclusion. For
instance, the abstract mentions that the plume migrates farther at a slope of 3° compared to a flat area.
However, this observation is only valid under specific flow field conditions. Plumes can still be observed at
distances of 4.5 km, 500 m, and 50 m. It would be beneficial to include information on the change rate of
plume concentration at these distances relative to the initial concentration value. Such data would provide
a more comprehensive understanding and should be carefully considered.

**Answer**

We agree that it would be very interesting to provide a change rate of plume concentration relative to the
source concentration, but source concentration is considered to be confidential information by DEME-GSR
and is also not given in the published paper from the near-field plume dispersion study²⁰. On the other
hand, source concentration is highly specific for vehicle and operational conditions, and what ultimately
matters for deep-sea life is not the rate of change relative to a maximum determined by human activity but
rather the excess relative to the natural baseline. We have thus chosen to compare excess measured
concentrations to these baseline concentrations. We have removed several specific numerical values from
the abstract and kept it more general as suggested by the reviewer (e.g., lines 22–28).

3. This manuscript exhibits characteristics more akin to a report rather than a scientific paper. The objective
of the experiment is not clearly stated, and the problem being addressed is not adequately reflected in the
manuscript. Such deficiencies are clearly unreasonable for a scientific paper.

**Answer**

The reporting style of the manuscript is directly associated with the comprehensive amounts and types of
data that we attempt to present. The scope of the paper is to report on plume monitoring methodology
and results, show how the most effective monitoring can take place and to deliver the physical data that
are required as a basis for data-driven environmental impact assessment. We have inserted clear
statements on the objective in the text in lines 55-56 and 496-499.

4. The length of the manuscript is excessive. Certain sections within the proposed methodology can be
omitted, while additional content can be appropriately included in the supplementary materials.
For instance, the description of the full water depth velocity profiler in Line 656 does not appear to be
utilized in this paper and is recommended for deletion. Similarly, the content spanning Line 765-782 would
be better suited for inclusion in the supplemental materials.

**Answer**

The Methods section is indeed long, as five different sensor groups have been used: MBES/SSS, Seafloor
camera, OBS, Water-column particle camera and ADCP. We think the methodology plays a vital role in the
quality of the results. Moreover, we believe that the methodology is especially interesting to marine
scientists involved in deep-sea monitoring, as there are currently no official standards or guidelines in data
post-processing. Bearing this in mind, we argue that it is better to present all sensor groups under the same
section (Methods) and not move ADCP sub-section (i.e. lines 765-782 in the old manuscript) into
supplementary materials. The Methods section is below the recommended limit of 3,000 words.
Concerning the comment for lines 655-656 (in the old manuscript), the full-depth Sound Velocity Profile
(SVP) ray tracing correction is the first and fundamental step in MBES post-processing. Without a corrected
SVP, the refraction of acoustic waves is not considered, and a wrong depth is estimated¹.

73 ^{R1} Lurton, X. An introduction to underwater acoustics, Springer-Verlag (2010).

5. In Line 164-167, the presence of gravity currents suggests the potential resuspension of seafloor
sediments, which necessitates further interpretation. It may be possible to calculate the critical shear stress
associated with this phenomenon.

**Answer**

The highest instantaneous current speeds recorded upon arrival of the plume at sensor locations
downslope of the collector test area were only slightly above 10 cm s⁻¹, which seems insufficient to erode
deep sea sediment¹. We cannot exclude, however, that 1) higher speeds in the head of the gravity current
were missed in our 5-minute interval recordings, and 2) higher current speeds may have occurred further
upslope than where the nearest sensors were located. But even if our observations provide no evidence of
erosive gravity currents, we think it is legitimate to suggest that gravity currents may become self-
reinforcing on steeper and more prolonged slopes. We added new text (lines 151-166), a new figure (Fig.
4; lines 219-226) and a link to online video in lines 510-511 of the revised manuscript, where we elaborate
on critical shear stress.

New text (lines 151-166):

"... The gravity current created sediment ripples (Fig. 3-4) with an NE-E current direction. The presence of
sediment ripples could imply the mobilization of additional surface sediments due to erosion³⁹. This would,
however, demand current velocities > 30 cm s⁻¹ as determined during experiments on sediment
resuspension in a large seawater flume using original sediments from the CCZ or as determined for deep-
sea sediments of similar particle composition^{29,40-41}. A more likely explanation is that the massive fallout of
aggregated sediments within the gravity flow⁴² (see Particle flocculation) combined with the unidirectional
currents in steeper seafloor parts resulted in mud ripples generated by large non-cohesive plume

aggregates. They behave hydraulically equivalent to fine sands⁴³⁻⁴⁴. Results from laboratory studies of
 aggregated plume sediments from the study site²³ using the approach of Thomsen & Gust (2000)⁴⁰, revealed
 that bedload transport generated ripples began to form at a critical shear stress of 0.04 N m^{-2} , and that
 resuspension of recently settled plume sediments started at 0.1 N m^{-2} (see Online Video Content). Multiple
 turbidity currents, originating from multiple lanes, could enhance the shape and dimensions of ripples.
 Larger sediment ripples were mapped at the NE end of the first mining strip (downslope direction), where
 the gravity current focused into the steeper NE running valley (Fig. 4). "

New figure (lines 219-226):

The new Fig. 4 and figure caption is presented below:

**Fig. 4 Formation of ripples downstream and downslope from the 1st mining strip.**

**a)** Orthophoto-mosaic of seafloor downslope from the end of the first mining strip before the mining trial. **b)** The same
 seafloor part after the mining trial. The formation of ripples heading towards the East (downslope direction) is clearly
 depicted. **c-d)** Topographic profiles based on the image-derived DEM after the mining trial. The depth of sediment erosion
 behind the nodule (c) and the ripple's lee side are shown (d). The ripple geometry (stoss side, crest, and lee side) is clearly
 depicted (d). The shape of ripple crests is sinuous to catenary, which is a typical bedform formed under a strong flow over
 cohesive sediments⁴⁵.

New online video (lines 510–511):

A video on resuspension and bedload transport of recently settled plume sediments at critical shear stress
 of 0.04 N m^{-2} is accessible here: <https://cloud.geomar.de/s/6qR8Df95n25fB6d>

6. In Line 317-318, the utilization of the coverage of polymetallic nodules following sediment cover as a
means to characterize the extent of plume cover is a commendable approach. However, this method
becomes impractical when the sediment completely covers the polymetallic nodules. Additionally,
accurately determining and calculating the cover area becomes challenging when the sediment cover area
is small.

**Answer**

To the best of our knowledge, no algorithm currently quantifies the thickness of sediment coverage on a
polymetallic nodule based directly on seafloor images. Some algorithms count the number, size and %
seafloor nodule coverage (i.e., how much seafloor is covered by nodules) and thus could be used to
estimate the spatial extent of the plume. We have added the method limitations (lines 742–762) in the
Methods section of the revised manuscript.

New text (lines 742–762):

"Such algorithms, including the one used here, have been used in the past to quantify nodules' spatial
distribution and support resource estimation studies^{32,84–87}. The same methodology has been applied to
quantify the spatial extent of sediment redeposition during an epibenthic sledge disturbance experiment
in the CCZ¹³. The difference in detectable % seafloor nodule coverage between the seafloor images before
and after the plume passed and (partially or totally) cover up the nodules shows the spatial extent and
degree of sediment redeposition. Such an approach works well when the redeposited sediment completely
covers the polymetallic nodules (i.e., 0 % detection) but not when there is a faint sediment cover (powder)
on top of the nodules before the additional sediment deposition^{87–89}. The latter is not the case in our study,
as shown in thousands of seafloor images acquired before the trial (e.g., Figure S16). The images were
obtained along two AUV transects in the SE direction ≈ 29 h before and ≈ 72 h after the mining trial. The
two transects have a similar length of ≈ 1400 m and a maximum distance of ≈ 85 m between them. Only
the images acquired in flying altitudes 5–7 m were analyzed as they have better quality characteristics (e.g.
brightness, contrast). Another limitation is that taking images from the exact same area of seafloor with an
AUV is difficult due to slight navigation uncertainties. Even state-of-the-art HUGIN class AUV cannot be
100% precise in XY direction during repeated surveys. In our case, we use images from survey lines that
cross each other at a slight angle. This was because of operational restrictions during the time of the AUV
dive, where two ROVs were used at the same time to recover equipment on the seafloor, and the AUV
needed to keep a safety distance, prohibiting flying exactly the same profile as before the plume settled."

In addition to Figure 10 we have added Fig. S16 (lines 108–116 in revised Supplementary Information), in
which the difference in detectable % seafloor nodule coverage is shown for pre- and post-trial seafloor
images in different distances.

Fig. S16 (lines 108–116 in revised Supplementary Information):

a		Pre-Impact	Post-Impact
Distance from mining lanes (m)	50		Distance from OBS (m)	0.15		
Maximum SPM (mg L ⁻¹)	264		
Pre-detectable coverage (%)	44		
Post-detectable coverage (%)	1.7		
b			Distance from mining lanes (m)	500		
Distance from OBS (m)	200		
Maximum SPM (mg L ⁻¹)	35		
Pre-detectable coverage (%)	27.5		
Post-detectable coverage (%)	10.5		Distance from mining lanes (m)	1000		
Distance from OBS (m)	160		
Maximum SPM (mg L ⁻¹)	28		
Pre-detectable coverage (%)	27.1		
Post-detectable coverage (%)	9.5		Distance from mining lanes (m)	1800		
Distance from OBS (m)	300		
Maximum SPM (mg L ⁻¹)	3.9		
Pre-detectable coverage (%)	6.5		
Post-detectable coverage (%)	20.1		

**Supplementary Figure 16: Comparison of seafloor images at successive distances from the impact site before and after**
**the mining trial.**

**a)** 50 m W from the impact site (next to NIOZ_PFM-06 platform). **b-d)** 500–1800 m SE from the impact site along the AUV
survey lines shown in Figure 10. **d)** The images are at the end of the survey lines outside of the zone of plume sediment
redeposition. The lower nodule coverage in the pre-impact image is due to the normal seafloor spatial variability in nodule
coverage, the pre-impact image was taken within a depression that naturally has lower nodule coverage. The lateral distance
from the nearest OBS and maximum concentration recorded are also provided.

7. In the Discussion from line 374 to 382, it was mentioned that flocculation occurring during plume
settlement leads to an increase in the median particle size from 12 µm to 470 µm. It is suggested that the
sedimentation rate of particles after flocculation should be included for a more comprehensive analysis.
Additionally, it is recommended to incorporate the basic physical parameters of surface sediments in the
study area, such as density, water content, pore ratio, and consolidation coefficient, into the paper.

**Answer**

Regarding median size: We apologize for our lack of clarity in presenting the median particle size and the
mistakes made in the corresponding text (e.g., the 0.03–0.47 mm statistic was not referred to D₅₀ but to
the range of measurements). We have reworked the data and presented it in detail in response to the
seventh point of Reviewer #2.

Regarding sedimentation rate:

We revised the manuscript accordingly, adding the following new text in lines 286–298:
"Although the settling velocity was not measured in situ, laboratory experiments with similar-sized
aggregated plume particles from the study area revealed settling rates of ≈ 100 m per day. Such rates
resulted in short residence times within a 2-3 m high plume of 30 – 45 minutes, with more than 80 % of all
aggregated particles settled by then⁴⁹. If these particles settle within the first 500 m away from the nodule
collector, the corresponding flow velocity of the gravity current would be ≈ 0.15 m s⁻¹ for the D₅₀ and ≈ 0.2
174 m s⁻¹ for larger particles D₅₀ - D₇₅. These values are close to the discharge buoyancy velocity of ≈ 0.21 m s⁻¹
mentioned in the near-field study²⁰. They also agree with the in-situ observations from the OBS sensor at
NIOZ_PFM-06, which registered the plume for the first time on 19.04.2021 at 19:25:00. Assuming an
average and constant speed of ≈ 0.2 m s⁻¹ for a period of 600 s, the nodule collector should have been \approx
120 m away from the platform at the time of sediment discharge. Plotting the nodule collector track
positions shows that on 19.04.2021 at 19:15:00, the collector was 124 m away from NIOZ_PFM-06 on a flat
seafloor."

Regarding surface sediments: Most physical parameters of surface sediments are published in the
contractor's (GSR) Environmental Impact Statement¹⁷, which we have cited in the manuscript. We do not
deem it necessary to include them again here.

8. From line 383 to 387, it is stated that the underflow becomes the driving force of the plume motion after
a distance of 500m. However, it is unclear what the driving force is before reaching this distance. If the
underflow is not the driving force, the reason behind the plume moving in the ES direction should be further
explored. The Discussion in this section lacks clarity in conveying the author's intended message.

**Answer**

We rewrote this part for better clarity. The new text (lines 370-379) is:

"Our results demonstrate that the sediments mobilized by the nodule collector generated a gravity current
that propagated perpendicular to the mining lanes to distances of up to 500 m downslope from the impact
site. Up to this distance, the seafloor slope determined the E-ward direction of the gravity current. The
ambient SE-directed crossflow did not deflect the gravity current from its downslope trajectory within the
first 500 m. At the base of the slope a stretch of flat seafloor ($< 1^\circ$) bounded towards E by a sill forms an
enclosed basin. The sill blocked the propagation of the gravity current further to the east and the plume
started spreading laterally (Fig. 5). Such flow refraction/diversion of gravity currents has also been shown
in other studies^{51–52}. From there onward, ambient currents determined the sediment plume's direction
towards SE, with finer particles still in suspension dominating the plume composition."

9. Line 716-751. The manuscript describes the preparation of suspended particle liquids with varying
concentrations using soil samples collected from the surface of the study area. These liquids were then
tested for turbidity in the laboratory to establish the relationship between NTU and mg/L. Typically, to
ensure the reliability of such tests, this relationship is established based on the results obtained from
seawater sampling at different CTD levels. It is important to understand the differences between the results
obtained in this study and those from in-situ testing. Furthermore, the experimental setup in this study
bears similarities to a previous experiment conducted by Munoz-Royo et al. (2022). The attachment in this
paper provides a detailed description of establishing the relationship between NTU and mg/L, which is
recommended as a reference.

Muñoz-Royo, C., Ouillon, R., Mousadik, S.E., Alford, M.H., Peacock, T., 2022. An in Situ Study of Abyssal
Turbidity-Current Sediment Plumes Generated by a Deep Seabed Polymetallic Nodule Mining Preprototype
Collector Vehicle. Science Advances.

**Answer**

The approach mentioned by the Reviewer, in which turbidity sensors lowered on a CTD-Rosette are
calibrated by reference to suspended particulate matter concentration determined in simultaneously
collected water samples, is described, for example, in Haalboom et al. (2021)⁴⁷. It works well for calibrating

sensors to naturally occurring suspended particulate matter in settings where water with different particle
concentrations is within reach of the CTD-Rosette, such as the submarine canyon described in Haalboom
et al. (2021)⁴⁷, where turbid water masses extend up to hundreds of meters above the seabed. The
approach is impracticable in a sediment plume monitoring exercise as described in the present manuscript
for two main reasons: 1) without information on the spreading of the plume in real-time, it is very hard to
determine when and where to deploy the CTD in order to hit the plume, and 2) in its initial stage the plume
extends only few meters above the seabed, closer to the seabed than what is generally considered the
minimum safe distance for lowering the CTD in the open ocean. The difficulty of sampling a mining plume
with CTD-Rosette is described in Haalboom et al. (2023)⁵³. Since the sediment plumes investigated in our
study were artificially produced by the mining machine stirring up fine-grained seabed sediment into the
bottom water, suspensions produced in the lab from seabed sediment and bottom water collected from
the same region provide a good analogue for calibrating sensors in the lab. We do not feel a need to refer
to the calibration method as described in Muñoz-Royo et al. (2022)²⁰. The principle of calibrating sensors
by reference to artificially prepared sediment suspensions has been described earlier by Guillen et al.
(2000)⁹⁰ and for CCZ sediment by Haalboom et al. (2022)⁵. The method description by Muñoz-Royo et al.
(2022)¹² reads rather cumbersome as different sets of sensors were calibrated in separate rounds, which
does not seem to be ideal. Nevertheless, it was still shown that the calibration relationship established was
the same in the second decimal digits for the fresh suspended and seafloor sediment samples (Figure S7)²⁰.
Their paper is cited within our manuscript.

The following sentences have been added in lines 810–816:

"Since the sediment plume investigated in our study was artificially produced by the mining machine stirring
up fine-grained seafloor sediment into the bottom water, suspensions produced in the lab from seabed
sediment and bottom water collected from the same region provide a good analogue for calibrating sensors
in the lab. The principle of calibrating sensors by reference to artificially prepared sediment suspensions
has been described earlier by Guillen et al. (2000)⁹⁰ and for CCZ sediment by Haalboom et al. (2022)¹²."

10. There are some minor errors in formatting and spelling in the manuscript, such as:

Line 192 "1800m" should be "1800 m".

Line 687 "absoluteseafloor" should be "absolute seafloor".

Line 790 62.33mm s-1, should be "62.33 mm s-1".

**Answer**

Line 192: Corrected (now line 237)

Line 687: Corrected (now line 733)

Line 790: Corrected (now in line 886)

Fig. 1. Only what is shown in figures b and c is marked. The section shown in Fig c is the a1-a2 line in Figure
249 d, which is difficult for readers to read in order.

**Answer**

Corrected. Please see our response in the figure comments from Reviewer #3.

Extended Data Fig. 2. The horizontal coordinate in the figure is not marked, although I know it is the date,
but it does not meet the specification of drawing.

**Answer**

Corrected. Please see our response to Reviewer #2.

**Reviewer #2 (Remarks to the Author):**

The manuscript describes plume monitoring during a pilot test of deep sea mining equipment. This study
is unique in its kind and provides a significant contribution to quantify the environmental impact that might
be resulting from seabed mining activities. It is highly appreciated that the monitoring campaign is
conducted independently, i.e. through government funding, supporting transparency and openness.
The authors provide extensive explanation and supporting evidence of how they have conducted their
research. The manuscript provides good insights that are complementary to the near field measurements
conducted by MIT. Together with the various sorts of data collected, the authors have managed to provide
a clear picture of what most likely has happened during these tests. In most cases, sufficient explanation
has been provided to prove or to philosophize what mechanisms or probable causes determining the main
outcomes of the pilot vehicle test. I would recommend that this manuscript will be published, after the
feedback has been addressed. This research is unique and of very high relevance towards responsible use
of the oceans and exploitation of the seabed.

**Answer**

We thank Reviewer #2 for the time and effort spent revising our manuscript. We welcome this very positive
evaluation and acknowledgement of this study's unique character, which was conducted independently
(the only modern in-situ deep-sea trial monitoring study that has been funded through government
funding until now). We have addressed the comments and concerns point-by-point.

General remarks:

1. The study only mentions the turbid flow resulting from the discharge of the excess of water and sediment
flowing through the collector vehicle. Although it is the main source of turbidity, the sediment kicked up
by the caterpillar tracks and the sediment spilled around the collector device can still be a significant
amount. Please reflect on this.

**Answer**

The Reviewer is right in pointing out that the plume produced by the collector vehicle does not only result
from the discharge of excess water and sediment at the rear of the vehicle. The sediment kicked up by the
caterpillar tracks and spilt around the collector also contributes to the total sediment resuspension and
plume concentration. According to published data, the collector's movement initiates a sediment flow of
$3 \pm 2 \text{ kg s}^{-1}$ when all pump heads are turned off²⁰. Another study also observed a distinct sediment plume
during tests of a different pre-prototype collector vehicle in the Alboran Sea, without the hydraulic
collector actually working⁵³.

Thus, two sentences have been revised as follows:

Lines 370–371: "Our results demonstrate that the sediments mobilized by the nodule collector generated
a gravity current that propagated perpendicular to the mining lanes to distances of up to 500 m downslope
from the impact site..."

Lines 67–69: "... Together with sediment stirred up by the locomotion on the seafloor, the initial input to
the plume during the trial was estimated to be $12 \pm 3 \text{ kg s}^{-1}$ of sediment²⁰.

2. Line 104/105: are all three strips driven from SW to NE? If so, would be good to reflect on this in the
Discussion, as currents moving downslope of new lanes might catch up existing currents. Catching up of
turbidity currents has not been thoroughly studied and thus this might be an interesting option to be
considered for mitigation of impact (potentially preventing the ripples).

**Answer**

The first and third strips were driven from SW to NE (extending towards the downslope direction). The
second strip was driven from NE to SW (opposite direction, extending towards the upslope direction). The
corresponding lines have been revised as follows:

Lines 98–100: "...The lanes were distributed over three strips (first strip: 55 lanes, second strip: 31 lanes,

third strip: 85 lanes; Fig. 1). The first and third strips were driven from SW to NE, whilst the second strip
was driven from NE to SW..."

Reviewer #2 made an excellent point here that it is readily resolvable. The turbidity current has a
time/distance-varying propagation speed, which is affected by the seabed topography, the orientation of
the mining track and the ambient bottom water currents. Unfortunately, we lack measurements on the
initial phase of the turbidity current development close to the collector before the plume moves
downslope. Higher speeds of the turbidity current are expected further upslope of our closest sensor
location. This is due to the limited number of sensor platforms that need to be deployed simultaneously
and especially due to the minimum safety distance to the nodule collector itself. We also miss the turbidity
current speed below 2 m altitude for all sensors apart from the 2 MHz ADCP on the platform (NIOZ_PFM-
07); this is because of the blanking distance of approx. 1- 2 m of ADCP sensors with lower frequencies. In
addition, the 5-minute measuring interval (this interval was selected to optimize the ADCP battery capacity)
means that we might have missed higher current speeds at the front of the gravity current. To approach
this question, we came up with the following calculations:

The nodule collector needed ≈ 540 s to move from the central point A of a mining lane to the central point
B of the next parallel line (route distance of ≈ 135 m, including the light bulb turn maneuver and nodule
dumping). The lateral distance between points A and B was ≈ 4 m. Assuming the initial turbidity current
speed of 0.2 m s^{-1} to be constant (please see our response to comment #7 from Reviewer #1), the turbidity
current needs 20 s to cover the distance of 4 m to the next mining lane. This time increases if we use the
mean current speed measured at the sensors closest to each mining strip when the plume was passing by:

Mining strip #1:

- • 0.06 m s^{-1} at 2.25 m above seafloor (NIOZ_PFM-03)
- • 0.07 m s^{-1} at 5.25 m above seafloor (NIOZ_PFM-03)

Mining strip #2:

- • 0.05 m s^{-1} at 2.25 m above seafloor (NIOZ_PFM-06)
- • 0.09 m s^{-1} at 4.75 m above seafloor (NIOZ_PFM-06)

Mining strip #3:

- • 0.05 m s^{-1} at 1.21 m above seafloor (NIOZ_PFM-07)
- • 0.05 m s^{-1} at 1.99 m above seafloor (NIOZ_PFM-07)

Based on these speeds, the turbidity current needs 44-80 s to cover the distance of 4 m to the next mining
lane. However, the nodule collector would be there only 540 s later. These calculations are corroborated by
the ROV footage, which shows the turbidity current entering a previously mined lane. The turbidity current
generated during the mining of the previously mined lane has already propagated further to the east
(downslope direction) and the water visibility returned to normal. We have added this footage in Fig. 3c.
The revised Figure 3 is in lines 201–218).

Although it seems unlikely turbidity currents from parallel lines merged in the initial stage of propagation,
we cannot exclude that this occurred later (e.g., at the base of the slope or exactly afterwards in the flat
seafloor). We think that the merging and the collision of turbidity currents generated by the mining
vehicle(s) is a field that should be investigated in more detail, especially when two or more nodule collectors
are mining simultaneously side-by-side in parallel lines. In this direction, dual-lock-exchange experiments
have been recently presented⁷⁵. We have also added the following text:

Lines 444–446: "Moreover, the merging of multiple gravity currents generated by the mining vehicle(s) is a
field that should be investigated in detail⁷⁵. Both could be of interest if one or more nodule collectors
operate continuously for weeks or months over the seafloor. "

3. Fig 2b3: Why is it that the depth along the centerline is taken as 0? If the profile is taken slightly longer,
then the 'virgin' bed outside the lane could be used as a reference of 0. That would also give a more fair

interpretation of the amount of deformation underneath the tracks (compaction + scooping), as well as
erosion depth caused by the collector device.

**Answer**

We have reworked Fig.2 and now the profile (b3) is taken slightly longer in both directions, showing the
erosion depth caused by the collector (as it is derived from seafloor images acquired after substantial
sediment redeposition within the mining site) and sideways sediment extrusion.

The revised Fig. 2 is in lines 122–137.

4. Lines 166-167: Is there any data collected that might provide insight on the propagation speed of the
current? If so, how do the current velocities relate to the erosion velocity of the sediment? Is there any
information available regarding the erosion velocity of the 'virgin' seabed and that of the freshly deposited
material. This might be relevant because to identify whether mitigation might benefit from longer time
intervals between neighbouring lanes.

**Answer**

This comment is similar to the fifth comment from Reviewer #1. Please see our response there.

5. Line 167-168: Based on the map in fig1, it might have been interesting to consider similar mapping from
area 3 to mooring NIOZ_PFM-03, to identify whether sediment ripples would have occurred parallel to the
lanes. Is there data collected?

**Answer**

The acquired seafloor images and generated orthophoto-mosaic do not extend until platform NIOZ_PFM-
03. The orthophoto-mosaic extends from mining strip #3 to mining strip #1. Within this extent, no sediment
ripples are oriented parallel to the mining lanes. Sediment ripples were found only perpendicular to the
mining strips #1 and #3 in the downslope direction. No sediment ripples were observed at the end of mining
strip #2 (flat seafloor). Covering an even larger area was not possible in the given time and the available
number of AUVs.

6. Line 171-172: Would this have been an effect of a single turbidity current, or is it caused by multiple
turbidity currents catching up on each other along the slope, originating from multiple lanes? Would you
eventually get a larger single gravity current, or multiple?

**Answer**

Lines 171–172 of the initially submitted manuscript described the absence of sediment ripples upslope of
all mining strips. Since ripples were observed only in the downslope parts, we infer that the turbidity current
generated behind the nodule collector is not strong enough to erode the seafloor and form ripples in the
upslope direction. The turbidity current propagation in the upslope direction was limited.

In the downslope direction, the turbidity current propagation has resulted in interaction with the mobile
substrate and the formation of ripples. (please see our response to the fifth comment of Reviewer #1).

Multiple turbidity currents catching up on each other along the slope, originating from multiple lanes, could
enhance the shape and dimensions of ripples. The last part of the question ("Would you eventually get a
larger single gravity current or multiple?"), has been addressed in general remark #2 of Reviewer #2).

7. Line 274: statistic of 0.03-0.47mm does not match with fig 5b. Based on what is visible in that figure,
$d_{50} < 0.06$ does not occur. Furthermore, I would suggest to include data of NIOZ+PFM-03 next to this as
well, to help interpret the particle number to concentration. Furthermore, the figures would probably
improve if a distinction would be made between when the plume passes by and where there is no to little
plume (particle count is very low, but PSD seems quite spiky in that interval). 5b shows a few peaks $> 400 \mu\text{m}$.
Is there any explanation why these spikes are observed? And the numbers on y-axis are not readable and
not properly positioned.

**Answer**

We apologize for our lack of clarity in presenting the abovementioned graphs and the mistakes made in the
corresponding text (e.g., the 0.03–0.47 mm statistic was not referred to D_{50} but to the data range).

We have reworked the and revised the manuscript, figures and supplementary information accordingly.

Lines 280–298:

"The PartiCam (JUB_PFM-01) showed that whereas the median particle size (D_{50}) of the top 5 cm of the
seafloor sediment is 0.012 mm¹⁷, the plume D_{50} increased up to 0.147 mm during the second and third
plume episodes (Fig. 8). Throughout the trial, which lasted longer than the minimum time needed to form
flocs with CCZ sediment in the lab^{23,48}, the D_{25} – D_{75} ranged from 0.061 mm to 0.313 mm (Fig. 8 and Table
S1). The largest particles were detected during the third plume episode, when the plume was produced
furthest away from the camera and exhibited the longest aggregation time (Fig. 8 and Table S1). Although
the settling velocity was not measured in situ, laboratory experiments with similar-sized aggregated plume
particles from the study area revealed settling rates of ≈ 100 m per day. Such rates resulted in short
residence times within a 2-3 m high plume of 30 – 45 minutes, with more than 80 % of all aggregated
particles settled by then⁴⁹. If these particles settle within the first 500 m away from the nodule collector,
the corresponding flow velocity of the gravity current would be ≈ 0.15 m s⁻¹ for the D_{50} and ≈ 0.2 m s⁻¹ for
larger particles D_{50} - D_{75} . These values are close to the discharge buoyancy velocity of ≈ 0.21 m s⁻¹ mentioned
in the near-field study²⁰. They also agree with the in-situ observations from the OBS sensor at NIOZ_PFM-
06, which registered the plume for the first time on 19.04.2021 at 19:25:00. Assuming an average and
constant speed of ≈ 0.2 m s⁻¹ for a period of 600 s, the nodule collector should have been ≈ 120 m away
from the platform at the time of sediment discharge. Plotting the nodule collector track positions shows
that on 19.04.2021 at 19:15:00, the collector was 124 m away from NIOZ_PFM-06 on a flat seafloor."

Lines 299–310:

**Figure 5. Particle flocculation was recorded in situ.**

a) Particle number records during the mining trial using the PartiCam data from the JUB_PFM-01 platform. Three successive
plume episodes were recorded (grey areas), corresponding to the three mining strips. **a1)** ROV image above the bulk of the
plume (4.7 m altitude) at the JUB_PFM-01 location. **a2)** ROV image within the bulk of the plume (4.3 m altitude) at the same

location 40 s later. The reduced visibility becomes obvious, with the sensor (Microprofiler)² hanging from the ROV arm at
 about 1 m from the ROV camera not being visible in a2. **a3–a4**) In-situ PartiCam images during the first day of the mining trial
 show a decrease in number of particles and an increase in particle size with time. The red scale bar corresponds to 0.27 mm.
 The minimum particle size that could be detected with that optical setup was 0.0153 mm with a pixel resolution of 0.0051
 424 mm. **b**) Particle size distribution during the mining trial. The black line corresponds to the D₅₀ particle size, and the pink area
 shows the D₂₅–D₇₅ particle size range. **b1**) Boxplots of the median particle size for each plume episode.

Lines 843–846 in Methods:

"...A rolling median (k=3, with k being the integer width of the rolling window) was applied to remove a few
 isolated outliers from the raw data and allow better visualization of the trend of increasing D₂₅, D₅₀, and
 D₇₅ during the mining trial. The zoo package⁹⁵ in R⁹⁶ was used. The raw data before and after the application
 of rolling median are shown in Fig. S18."

Lines 92–93 in Supplementary Information:

**Supplementary Table 1: Summary statistics of the particle size (D₂₅, D₅₀, D₇₅) for each plume episode.**

1st Plume Episode	Min.	1st Qu.	Median	Mean	3rd Qu.	Max.
D ₂₅ (raw)	59.0	69.7	73.0	74.7	77.0	101.0
D ₂₅ (roll median)	61.0	70.0	73.0	74.2	77.0	97.0
D ₅₀ (raw)	86.0	99.7	107.0	110.5	118.0	150.00
D ₅₀ (roll median)	86.0	102.0	108.0	109.3	117.0	145.00
D ₇₅ (raw)	115.0	139.0	148.5	164.9	175.0	473.0
D ₇₅ (roll median)	124.00	143.2	149.5	155.7	170.0	202.0
2nd Plume Episode	Min.	1st Qu.	Median	Mean	3rd Qu.	Max.
D ₂₅ (raw)	62.0	90.5	104.0	100.6	114.0	129.0
D ₂₅ (roll median)	68.0	92.5	106.0	101.0	114.0	122.0
D ₅₀ (raw)	78.0	129.2	147.5	142.9	159.8	187.0
D ₅₀ (roll median)	97.0	130.5	148.5	143.5	159.0	169.0
D ₇₅ (raw)	93.0	172.5	195.0	190.6	209.8	267.0
D ₇₅ (roll median)	130.0	174.0	195.0	189.8	209.0	246.0
3rd Plume Episode	Min.	1st Qu.	Median	Mean	3rd Qu.	Max.
D ₂₅ (raw)	66.0	82.0	97.0	98.6	116.0	152.0
D ₂₅ (roll median)	68.0	82.0	101.0	98.5	116.0	140.0
D ₅₀ (raw)	95.0	123.0	146.5	144.6	165.5	218.0
D ₅₀ (roll median)	100.0	123.0	147.0	142.0	164.0	200.0
D ₇₅ (raw)	130	178.5	200.0	210.0	229.0	487.0
D ₇₅ (roll median)	140	177.0	200.0	199.6	218.0	313.0

**Supplementary Figure 18: Few isolated outliers are observed in raw data.**

**a)** Time series of particle size (raw data) during the three plume episodes, which are shown in grey. The
 black line is the D_{50} , and the red area shows the upper D_{75} and lower D_{25} of the particle size distribution. **b)**
 Same as a, after applying a rolling median ($k=3$, with k being the integer width of the rolling window). **c)**
 Boxplots of the particle size for each distinct plume episode. **d)** Same as c, after applying a rolling median
 ($k=3$).

The application of a running median removes a few outliers, allowing for better visualization of the
 increasing trend of D_{25} , D_{50} , and D_{75} during the mining trial. We think these outliers are isolated flocculants
 of larger size that are not representative of the particle size distribution of the plume. The small peaks
 between the second and third episodes are attributed to residual particles within the camera field of view.
 For consistency and to avoid figure repetition, we prefer to keep the SPM concentration plot from JUB_PFM-
 01 and NIOZ_PFM-03 in Fig. S2 (lines 25–30 in revised Supplementary Information).

8. lines 304-308: Suggest to include that this is a conservative calculation. As the vehicle moves along the
 mining strip, it disturbs the recently settled materials from previous lanes. Whether this influences the
 source term (amount of sediment entrained by the collector device), probably no such detailed
 measurements of the source term exist.

**Answer**

This was also commented on point 5 of Reviewer #1 (please see our response there).

9. Lines 317-319: Although the qualitative trend of blanketing is clear, it would be good to explain how this
 is defined. What metric is used to assess the severity of blanketing? And what criteria are used to
 distinguish between faint, medium, thick and blanketing?

**Answer**

The method used to assess the severity of blanketing is explained in detail in our response to the sixth
 comment of Reviewer #1. It is now presented in detail within lines 739–763 of the revised manuscript.

The criteria used to distinguish between faint, medium, and thick are described in lines 764–773 of the
revised manuscript, which reads as:

"Parallel to detectable % seafloor nodule coverage, qualitative criteria were used to distinguish the degree
of sediment redeposition as follows. Blanketing: Nodules are entirely covered by redeposited sediments.
The open spaces between neighbouring nodules are filled by redeposited sediment. The nodule shape is
not distinguishable, and only the nodules' top part, protruding slightly from the seafloor, is still visible. The
AUV seafloor photos are yellowish, and the nodules' dark (black) colour is not visible anymore. Thick:
Nodules are entirely covered by redeposited sediments, but their shape is distinguishable. The nodules'
dark colour is still not visible. Medium: Nodules are partly covered by redeposited sediments. The nodules'
dark colour is still discernable. Thin: Nodules are partly covered by redeposited sediments. Their dark
colour is slightly shaded by the brighter sediment."

10. Discussion: Suggest to add a statement in the Discussion that the observed suspended material is more
than just that originating from the discharge of the mining vehicle. The caterpillar tracks bring sediment in
suspension and around the collector heads not all suspended sediment will be entrained into the system.
These sources of suspended sediment behave differently compared to the discharge. reference 2, showed
clear indications that the turbidity originating from driving only is still quite significant.

**Answer**

We agree with this suggestion (please see our response in general remark #1 of Reviewer #2), and we have
added a new statement in the Discussion of the revised manuscript, as follows:

Lines 440–444: " Additionally, the sediment in suspension around the collector heads and caterpillar tracks
should be considered. This resuspended sediment is not entrained into the nodule collector, having
different characteristics (e.g., sediment size distribution) and behaviour than the discharge plume. It
contributes to the total plume concentration within the collector impact site, but it is not clear if it is
integrated into the gravity current and far-field plume transportation."

11. Line 388-390: your observation shows a higher concentration than that is reported by the MIT study.
Could there be a plausible explanation?

**Answer**

The sensor that recorded the highest suspended sediment concentration of 264 mg L^{-1} was located about
50 m down-current from the nearest point where Patania II turned after running a straight lane and dumped
its load of collected nodules (NIOZ_PFM-06 platform). The sediment plume created in this maneuver may
have been simply more substantial (or captured in an earlier and more concentrated stage) than the plume
produced during normal operation on a straight line, such as in the selfie experiments described in the MIT
study. Another possible explanation is that during the trial presented here, the nodule collector picks up
plenty of freshly resettled material as well (and this is more comparable to a real mining scenario). That is
not the case for the selfies tests described in MIT study. In this direction, we have added the following
paragraph in Discussion:

Lines 421–428: "Whereas observations made at short range from the nodule collector showed that the
plume stayed mostly below 2 m altitude²⁰, our results demonstrate that it expanded in height, albeit at
much lower concentration, while spreading from the source. Similar observations have been reported
during deep-sea mining pre-prototype collector tests in shallow water⁵³ and benthic disturbance
experiments⁵⁴. In addition, the maximum concentration recorded here is higher than reported from initial
testing of the same nodule collector²⁰. These differences show that more in-situ trials must be closely
monitored, particularly with larger nodule collectors or nodule collectors that use different technology."

12. Research outcomes: Fully agree on the two main directions. However, regarding point 1, there is a
significant difference between the work that has been done in this study, scientific validation and scientific
measuring of the plume. For industrial applications, such a monitoring strategy is not fit for purpose.
Together with the thresholds discussion, monitoring strategy will have to be adapted. It is likely that

thresholds of e.g. max SPM at distance x will be used to manage a mining activity (environmental pressure
or direct measurement). However, in the light of uncertainty of the thresholds and cumulative impacts, it
is essential that technologies will be developed to enable (more) rapid assessment of the status of the
habitat (environmental impact, typically indirect measurement).

**Answer**

We agree with these comments. This study aimed, among others, to examine the applicability of currently
available sensors to monitor plumes and mining imprints, and identify gaps for improvement. We have
reworked the Discussion to highlight this need:

Lines 476–479: "Using such data and numerical models linked to 4D visualization capabilities that enable
extracting, exploring, and visualizing quantitative and semantic information in near-real time would allow
for meaningful and tailored adaptive environmental monitoring during future mining activities."

Lines 482–484: "Acoustic communication, small-cabled sensor networks, or repeated upload of AUV data
at central data nodes can already be utilized today."

Detailed remarks:

1. lines 71-75: A few questions, 1) is there quantitative info available regarding the quantity of water flow,
and thus sediment concentration of the discharged mixture? 2) What is the vehicle velocity? The mixture
is discharged from a moving source and thus concentration and vehicle velocity can be of influence on how
dense the initial gravity current will be.

**Answer**

1) To the best of our knowledge, there is no published and available information regarding the sediment
concentration of the discharged mixture. The only published information is the discharged height of 2.5–
3.2 m above the seafloor and discharge rate of $12 \pm 3 \text{ kg s}^{-1}$ of sediment²⁰, which has been included in lines
68–69 in the revised manuscript. This is an excellent point from Reviewer #2, as such information is vital,
particularly for the 4D plume dispersion modelling. We encourage publishing the sediment concentration
at the source, if possible, but this needs to come from company operating the collector vehicle and such
information has not been provided to us.

2) The vehicle velocity was not constant. It was moving along the 50-m-long mining lanes (during which the
head pumps were on) with 0.25 m s^{-1} . This value is based on the time recorded on USBL data at the start
and end of each mining lane, and it is close to the published data ($\approx 0.28 \text{ m s}^{-1}$) from a similar experiment²⁰.
The light bulb turn maneuver ($\approx 60 \text{ m}$) and nodule dumping were done in $\approx 10 \text{ min}$, which shows an average
speed of 0.1 m s^{-1} during this maneuver.

2. fig 1c/d and caption: caption states 1c top and 1c bottom, but 1c bottom is 1d

**Answer**

Corrected. The revised Fig. 1 is in lines 109–121.

3. fig 2, b2) suggest to include where the line starts and where it ends for ease of interpreting b3.
Furthermore, probably it is safe to assume that the bed imprint is symmetric along the centerline of Patania
II. Then I'd suggest to let the line start at the middle between the tracks.

**Answer**

Corrected. We have reworked Fig.2 and the profile (b3). We have marked the start and end of the profile
(d/d'). The profile starts almost from the middle between tracks (d) and extends to the seafloor area after
the sediment humps (d'). A more extended profile was not preferred as it demands a change of the
horizontal scale in images b1 to b3. The details in images b1 and b2 are particularly essential to show the
unmined but blanketed nodules next to the mining tracks.

The revised Fig. 2 is now in lines 122–137.

4. line 155: Moved slowly back and forth. It is not clear to me what is meant here. Can you reword? Does
it mean that the front was not always at a similar velocity, possibly due to discharge conditions (direction,
velocity and action of vehicle) and deviations in slope along the trajectory?

**Answer**
With the phrase "...and slowly moved back and forth...", we meant a gentle oscillation backwards and
forwards. Nevertheless, the net movement was forward. Line 155 of the old manuscript has been revised:
Lines 144–145: "Although a gentle oscillation was observed, the net movement was downslope in the form
of a gravity current (Fig. 3)."

5. caption fig 4, lines 259-260: it appears as if the effect of the flotation body is explained twice, but at
different positions

**Answer**

Corrected. The repetition has been deleted, and the new sentence is:

Lines 265–267: "... Although the background noise between 18-22 m above the seafloor (caused by a
flotation body on the platform) dominates the signal, three distinct plume episodes are still visible."

6. lines 291-294: Please rewrite, without reading the methods section, it is very difficult to understand how
this needs to be interpreted.

**Answer**

We have rewritten the abovementioned lines as follows:

Lines 333–342: "A simple estimation of the sediment mass transported at 500 m SE from the impact site
was attempted using the MBES WCI data and the OBS data from NIOZ_PFM-04 and RBIS_PFM-01 (see
Methods). During a two-hour mining activity in the first strip, 11.2 – 12.3 % of discharged sediment mass
passed through the lower 2 m and an additional 0.81 – 1.02 % of sediment mass was estimated to pass at
5–6 m altitude (Fig. S17). This estimation sets a lower limit, with the actual percentage expected to be
higher as the locations of platforms are on the southern edge along the recorded MBES WCI data, where
lower concentrations were observed (Fig. 3). Moreover, the OBS sensors at those platforms measured the
concentration at 1 m altitude. However, a portion of the plume had already ascended to higher altitudes."

7. Line 677: this is the first and only time DSM is used. Please explain.

**Answer**

Corrected (DSM=DeepSea Monitoring group) – line 722 in revised manuscript.

8. Line 688: add space to absolute seafloor

**Answer**

Corrected – line 733 in revised manuscript

9. Line 692: Please provide the name of the software

**Answer**

The software is called Mangan Analyzer. The sentence was revised to:

Lines 739–740: " The percentage of nodule seafloor coverage was quantified using the saltation GmbH &
Co. KG proprietary software Mangan Analyzer, fine-tuned for this explicit dataset."

10. extended data fig 2: Add label to x-axis. It is done in all other figures except this one.

**Answer**

Corrected. There is now a horizontal coordinate. In addition to the x-label, the symbol \leq has been added
before the distances, as those refer to the maximum distances. The revised figure (Fig. S2) is now in revised
Supplementary Information (lines 26–30).

11. extended data fig 6: suggest to include larger scale, or enable stronger contrast. When printing this
would not be visible

**Answer**

The former extended data fig 6 has been integrated into Fig. 5 and a stronger contrast has been used.
(please see our response to comment #7 from Reviewer #2). – lines 299–310 in revised manuscript.

**Reviewer #3 (Remarks to the Author):**

Review of Deep-sea polymetallic nodules mining imprints and plume monitored in situ- Isobel Yeo, NOC

This is an interesting paper and I enjoyed reading it. The authors are presenting data on one of the most
realistic mining collection trials to date, and as such the data presented here are of interest to both
scientific and industry audiences. Their study is timely and useful and I look forward to seeing it published
in Nature Communications. I have a few broad comments and a number of smaller detailed ones, none of
which I think should present the authors with too many problems to address. Also, I don't know if I am
allowed to comment on this but I shall anyway, the author list very male. Obviously, this now has to reflect
the people who have done the work, but I would encourage this group to explore why so few women were
involved in the study. Overall though I think this is a fascinating piece of work and I expect the results to be
used widely in the coming years as seafloor exploitation guidelines are developed.

**Answer**

We thank Dr. Isobel Yeo (Reviewer #3) for the time and effort she spent revising our manuscript. We
welcome the very positive judgment. We appreciate the concern raised on gender representation in the
authors list. We would like to state that all authors actively support equal gender participation in scientific
research. As mentioned by Reviewer #3, the author list reflects the scientists who have participated in this
specific manuscript. This author list does not reflect the total gender distribution within the MiningImpact2
project. We are only a small group of the entire transdisciplinary project (\approx 100 scientists, women and men
at a ratio of almost 1:1, distributed across three cross-cutting themes and five working packages,
representing a multitude of nationalities -also including developing countries- working at 29 institutes in 9
European countries). We have addressed the remaining comments and concerns point-by-point.

General comments:

1. Deep Sea exploitation is an emotive issue, and the authors are to be commended on this work, which
provides useful information for understanding the impacts of nodule mining on marine ecosystems.
However, there are a few times in which I think the authors underplay work that has been done (e.g. in line
53). I think paper this would benefit from a brief synthesis of some of these excellent past studies and an
identification of the major research and knowledge gaps, one of which, plume dispersion, this paper
addresses. Perhaps by expanding the text in lines 55-58. There are also a few parts of the paper that appear
to have an anti-mining stance. While I sympathize with the author's positions here, in my opinion it is not
our role to apply our own feelings to scientific results (which in this case are extremely useful). It is up to
the authors, but I would dial this back slightly.

**Answer**

We agree that science should be objective and neutral. This is our aim, and we thank Reviewer #3 for
pointing it out. The choice of strong forms of nouns, verbs and particularly adjectives is intended to
emphasize the content and not to state a pro-mining or anti-mining opinion. We have accepted all language
suggestions removed or replaced strong forms of nouns and adjectives.

We think a brief synthesis of past studies will disorient the readers from the main point, which is the
detailed description and analysis of this world's first deep-sea trial of a pre-prototype polymetallic nodule
collector. Moreover, it will expand the length of the manuscript further. The past studies and major
knowledge gaps are identified in lines 53–56 of the revised manuscript:

"However, our knowledge of the monitoring technology needed, plume dispersion, sediment redeposition
and likely impacts of deep-sea mining activities is still incomplete due to the novelty of this industry and
the complexity of acquiring data on the resilience of deep-sea life to such disturbances⁷. Valuable insights
have been obtained from mining trials and benthic impact experiments carried out since the 1970s,
although those disturbances were of different nature, spatial scale and intensity than those expected from
industrial mining technologies^{8–16}. Herein, the most detailed, up-to-date in situ view of deep-sea plume
monitoring using hydroacoustic and optic sensors is presented. "

In particular:

• Lines 16/17 in the Abstract – maybe rephrase to, "It is likely that Deep-sea mining of polymetallic
nodules may start without finalized mining regulations." Or something equally unequivocally true

**Answer**

Corrected. The line has been removed.

Line 52 -I wouldn't say "dire need", I'd specifically state the research gaps that exist and need to
be addressed

**Answer**

Corrected. The word dire was deleted.

Lines 473 – 475 – I don't think you can say "race to the bottom"

**Answer**

Corrected. The sentence has been deleted.

2. There is no reference in the main text to baseline values although I think these are alluded to at various
points. Was this data collected? It is briefly mentioned that these measurements were in the hours before
and after the experiment. If this is the case, I think they may be missing normal variability, and this
limitation should be stated clearly.

**Answer**

This is a good point. It is worth mentioning that published values of concentrations of particulate suspended
matter in near-bottom waters (≤ 5 m above the seafloor), obtained by filtration of bottom water samples,
are surprisingly scarce for the CCZ region (see our response in second comment from Reviewer #1). Previous
studies in the GSR and BGR contract areas show close-to-zero background turbidity values near the seabed:
0.02 FTU for the GSR area¹⁷ and 0.08 FTU for the BGR contract area^{12,25,30}. Based on long-term observations
within the BGR area (unfortunately no long-term observations exist for the GSR contract area), the bottom
water turbidity values were stable even during the time of an eddy passage (with increased bottom current
velocities (with peaks of up to 20 cm s⁻¹)³⁰. The low number of long-term turbidity observations cannot
ensure the absence of extreme events (e.g., extended periods with bottom current velocities of 17-20 cm
s⁻¹ that could initiate sediment resuspension and consequent increase in turbidity values^{30,41,74}) and
supports the idea for having permanent seafloor observatories installed. The following lines were added:
Lines 436–438: "...In this direction, the deployment of long-term seafloor observatories at successive
distances from the mining site could enrich our knowledge on far-field plume dispersion, ...").

Within the limited time frame of our monitoring we found only very low turbidity values in the undisturbed
near-bottom water, in sensors mounted on a CTD-Rosette, seabed sensor platforms and AUV (which
ensured also the good spatial coverage of our turbidity data). This was visually confirmed by the excellent
underwater visibility outside the plume. It is up to the contractors to demonstrate if there is any natural
turbidity variability of importance in near-bottom waters. To state clearly this limitation, we have done the
followings modifications in the revised manuscript:

Lines 89-91: "All OBS sensors and all acoustic doppler current profilers (ADCPs) measured ambient
concentration and background acoustic backscatter intensity before the trial during and directly after the
trial, respectively."

Lines 140-142: "This was four orders of magnitude above the measured background level of 0.02–0.03 mg
L⁻¹, but returned to ambient values after 14 h (Fig. S2–S3)."

3. I think there are two separate things that are important for monitoring impacts, the actual
measurements of the physical manifestations of mining activity, which is addressed by this study and which
will not vary too much between different sites, and then the threshold values for species that make up
ecosystems, which will be essential for understanding the actual impact activity has. These will vary
between sites. This is mentioned towards the end of the Discussion, but I think it might be worth clearly
separating these two out and describing this in the Introduction.

**Answer**

Indeed, this manuscript shows how the monitoring could be done based on the available technology and
highlights the need for further technological improvements (e.g., in-situ measurement of settling velocity,
large-scale quantification of sediment redeposition based on seafloor images). Understanding the current
technological solutions and their limitations is a first important step to define thresholds in the future: how
far from the source and how high/low can the available hydroacoustic and optic sensors measure the plume
SPM concentration? Where should platforms and long-term observatories be placed on the seafloor?
Comparative studies (undergoing research by the MI2 consortium) between seafloor parts with higher and
lower sediment redeposition will hopefully enlighten the impacts on deep-sea benthic fauna, helping to
establish such thresholds. This information could be used to prioritize mining in flat areas where turbidity
currents spread over shorter distances. Our findings show that plume models should consider sediment
flocculation, high-resolution bathymetry (probably the most neglected aspect until now) and the formation
of gravity currents. This is particularly important as such models would support defining locations for long-
distance and long-term seafloor observatories that monitor the deep-sea benthic organisms and plume
thresholds. This is now clearly stated in Discussion (Lines 488–495) and Introduction (Lines 49–52):
Lines 488–495: "Secondly, the most difficult challenge is to define quantitative tolerance threshold levels of
representative deep-sea biota to plume concentrations and thickness of sediment redeposition as a
measure to protect the wide range of benthic life inhabiting the seafloor areas targeted by mining. What
tolerance limits do deep-sea fauna have for suspended and redeposited sediment of certain particle size
distribution and chemical composition, and for what time of exposure, before suffering "Serious Harm"^{5,6}?
Relating impacts on benthic life observed after the trial to different levels of exposure to suspended and
redeposited sediment, as derived from our study, is a first step in that direction but requires time."
Lines 49–52: "However, our knowledge of the monitoring technology needed, plume dispersion, sediment
redeposition and likely impacts of deep-sea mining activities is still incomplete due to the novelty of this
industry and the complexity of acquiring data on the resilience of deep-sea life to such disturbances⁷."

4. There is no sediment characterization data presented here other than measurement made with the
PartiCam. I don't know if sediment traps were deployed, it seems likely that they would have been,
although they are not mentioned in the partner study either? Sediment characteristics are important for
understanding the ability of say, certain frequencies of ADCP, to detect them, as the authors say. If there
are no sediment characterization measurements of the actual plume material then I think this needs to be
made clearer, because while they have calibrated their OBS the material used in this calibration is probably
similar to but not identical to that kicked up in the collector experiment (I don't think they give the location
of the sample used for calibration relative to this study but I may have missed it) and therefore their
concentration values may not be perfect if it has a different particle size/shape distribution. If, on the other
hand they have samples of the actual plume material this data should be included.

**Answer**

The Reviewer #3 is right in pointing at the importance of particle size distribution in suspended matter as
determining the intensity of acoustic and optical backscatter. But whereas the particle size distribution in
disaggregated sediment samples can be routinely determined in the lab by established methods such as
laser diffraction particle size analysis, the particle size distribution of aggregated sediment as it occurs in
the plume can only be assessed in-situ, for example by particle cameras (other methods based on laser
diffraction and multi-frequency acoustic backscatter are currently under development). Sampling of plume
material for ex-situ analysis on board the ship or in onshore labs, as the reviewer seems to suggest,
inevitably changes the (aggregated) particle characteristics. For that reason, we made no attempt to
perform particle size analysis on suspended matter from the few CTD-Rosette water samples that could be
collected from the plume (the difficulty of sampling the plume with CTD-Rosette is explained in our answer
to the ninth comment of Reviewer #1), or in sediment trap samples². The surface sediment and deep water
used for making the calibration suspensions was taken from Stations IP21-070² and IP21-072², located
about 500 m E of the test area in the German exploration contract area². Despite the distance, the median
particle size of 17 µm measured in the used sediment suspension is within the range of median particle

sizes of surface sediments reported from the Belgian exploration contract area¹⁷. In a follow-up paper we
will explore how optical and acoustic backscatter response changed along and across the plume in relation
to particle size distribution. This topic requires more extensive explanation than can be accommodated in
the present manuscript, which deliberately focuses on relatively unambiguous observations regarding the
spatial and temporal occurrence of the plume and plume deposits.

5. There are a few places where I feel like the authors could have gone further with the data they have –
particularly as they are aiming at such a high-impact journal. The data here is mostly observational, and
very useful, but I think more could be done. For example, I don't think they have attempted ADCP inversion
or calibration, which would be of interest given they are using very high-frequency ADCPs (and I
congratulate them on deploying the correct equipment to measure a fine particle plume!). ADCP data is
likely to play a big role in monitoring seafloor disturbance going forward and it would be interesting to see
how successfully this data can be used to replicate their other results.

**Answer**

We welcome the positive comment on deploying the correct equipment to measure a fine particle plume.
We completely agree with the Reviewer that the use of acoustic backscatter recorded with ADCPs for
determining suspended sediment concentrations in the plume deserves further exploration, and we
certainly intend to do this in a follow-up paper; preliminary results have been presented at the EGU general
assembly 2023^{R2}. However, since we have no suspended sediment concentration data to calibrate the
acoustic backscatter records other than the data from co-located optical backscatter sensors, the
acoustically derived suspended sediment concentrations will largely reflect the values obtained with the
optical sensors. The added value of the acoustic profiles is that they reach a few meters (2 MHz ADCPs) or
tens of meters (300 kHz ADCPs) above the seafloor giving indications about the vertical spread of the
sediment plume. The authors have presented the relationship between ADCP acoustic backscatter data to
SPM concentration in older studies, including a deep-sea benthic disturbance experiment^{12,47}. We agree
that ADCPs would play an important role in monitoring plume concentration and height in the water column
as they have several advantages compared to other sensors used in this study: well-established knowledge
of the operation principles and data analysis, long-term battery life, short- and long-range measurements as
well as abilities for ADCP data inversion and calibration.

770 ^{R2} Diaz, M., et al. Monitoring flocculation during a deep-sea mining test in the Clarion-Clipperton Zone, eastern equatorial
Pacific Ocean, EGU General Assembly 2023, Vienna, Austria, 24–28 Apr, (2023).

In line 446 the authors refer to adaptive spatio-temporal modelling, which I agree is a challenge. I wondered
why the authors have not attempted to model the plume they measured. They reference Spearman et al.
(2018) which presents a model output for a smaller plume experiment, and I think a similar simulation
would be useful here. They could compare their modelled plume with the sediment redistribution and it
would also allow the authors to vary conditions to predict plume response, addressing the comment they
make in lines 406-407 about extrapolation of these results.

**Answer**

Our wish is to keep this paper a 100% data-driven approach based only on sensor measurements. As
elaborated in a previous comment (comment no. 3, Reviewer #3), it is important to present and understand
the current technological solutions and their limitations related to deep-sea plume monitoring. We fully
agree that the presented data is an excellent source for developing and validating a better/updated plume
model. The authors are currently working on such a model incorporating the information gathered from
the in-situ trials in the GSR and BGR contract area². However, the inclusion of the modelling approach in
this paper is beyond the scope of this manuscript for the following reasons:

a) A model presentation would divert the focus of the paper, which we would like to be the in-situ
trial, the monitoring methods and sensors used to monitor the seafloor activities – probably the
most holistic approach presented publicly until now.

- b) The main results of this study (e.g., slope and gravity flow, flocculation, rate of concentration decay
with distance and altitude) provide the information needed to build a representative model.
c) Using a model introduces additional uncertainties in interpreting the results.
792 d) The presentation, analysis, and discussion of such a model would substantially increase the
793 manuscript's length.

6. The figures are very data rich but, for this journal, I think need some improvement. I've given more
detailed notes below.

**Answer**

We welcome this comment and the suggestions made to improve the figures. We have revised the figures
accordingly to the suggestions provided.

Specific comments:

Line 19 – lose "scientifically", independent -> independently

**Answer**

Corrected. – now in line 18

Line 22- Has this been observed for other gravity currents? Obviously not for the abstract but interesting
to know if this is the first observation of this.

**Answer**

The trials in 2021 where the first in-situ deep-sea mining trial for polymetallic nodules on a sloping seafloor.
Thus, there is no other similar published observation regarding turbidity currents generated by a nodule
collector. A previous study has hypothesized that the presence of slope across the nodule collector mining
lane could increase the turbidity current run-out length²⁰, which is shown here for the first time. The
seafloor morphology complicates the plume spreading as it could both increase in length (e.g., slopes) or
block the turbidity current propagation (e.g., presence of a sill). The following text was added:

Lines 372–379: "...the seafloor slope determined the E-ward direction of the gravity current. The ambient
SE-directed crossflow did not deflect the gravity current from its downslope trajectory within the first 500
814 m. At the base of the slope a stretch of flat seafloor (< 1°) bounded towards E by a sill forms an enclosed
basin. The sill blocked the propagation of the gravity current further to the east and the plume started
spreading laterally (Fig. 5). Such flow refraction/diversion of gravity currents has also been shown in other
studies^{51–52}. From there onward, ambient currents determined the sediment plume's direction towards SE,
with finer particles still in suspension dominating the plume composition. "

Regarding the distance of 500 m, diluted plumes of lower concentration than in our study have been shown
to create turbidity currents that self-accelerated over such distance^{R3}.

821 ^{R3} Hage, S. et al. Direct monitoring reveals initiation of turbidity currents from extremely dilute river plumes. *Geophysical
Research Letters* **46**, 11310–11320 (2019).

Line 23 – Further than -> Beyond

**Answer**

Corrected. The new sentence reads: "...Beyond 500 m, ...". – now in line 21

Line 23/24 – A bit confused here – are the sediments beyond 500m not part of a gravity current?

**Answer**

We have rephrased it to:

Lines 19-22: "Findings show that the gravity current formed behind the nodule collector is channeled
through steeper seafloor sections (> 3° slope) and travelled 500 m downslope. Beyond 500 m, the prevailing
bottom currents dominated the sediment dispersion up to the end of monitoring area (4.5 km distance)."

Line 31 – I'm not sure this has been shown really – the sediment redeposition is qualitative. I would reword
this sentence.

**Answer**

Considering the reworked data, it has been revised to:

Lines 30-33: "For the first time at a water depth of 4500 m, a mm-scale photogrammetric seafloor
reconstruction allowed quantitative estimates of the thickness of the redeposited sediment next to mining
lanes (≈ 3 cm) and the minimum erosional depth (≈ 5 cm) as measured after sediment redeposition. "

Line 41 – reference needed here, probably HALBACH P. and OZKARA M. (1979) Morphological and
geochemical classification of deep-sea ferromanganese nodules and its genetical interpretation. In Proc.
Int. Colloq. C.N.R.S. No. 289, Sur la Genese des Nodules de Manganese (cd. C. LALOU), pp. 77-88. And/or
HALBACH P., SCHERHAG C., HEBISCH U. and MARCHIG V.(1981) Geochemical and mineralogical control of
different genetic types of deepsea nodules from the Pacific Ocean. Mineral. Deposita 16, 59-84.

**Answer**

Corrected. The reference has been added – line 43 in revised manuscript.

Line 52 – lose "dire"

**Answer**

Corrected. See previous comment for line 52 from the same Reviewer.

Lines 90-93 – How were the locations for the monitoring selected?

**Answer**

Lines 90-93 in the initial submitted manuscript referred to the AUV data collection at different altitudes and
distances from the collector impact site. This was the first time an AUV was mobilized to monitor the plume
dispersion during a mining trial; no previous experience or guidelines exist. Thus, the survey design included
several, mostly safety-related considerations:

- a) Minimum allowed AUV flying altitude.
- b) Maximum distance of the AUV to the surface vessel (allowing constant acoustic data
communication)
- c) Minimum allowed distance from the collector impact site during mining trials
- 859 d) Minimum allowed distance from ROV operations
- e) Maximum operation time given by the AUV battery capacity depending on the AUV sensor payload,
settings and speed.
- f) Need to generate high-resolution MBES bathymetry.
- 863 g) Bottom current directions.
- 864 h) Location of installed seafloor platforms
- i) Experience gained from older disturbance experiments

Line 97 – If these are the only background measurements, they miss quite a lot of possible natural variability

**Answer**

This was also commented in comment 2 of Reviewer #3 (please see our response there).

98 – Is this a name or should there be a space in MiningImpact?

**Answer**

The name of the project is without space. Please see: <https://miningimpact.geomar.de/de>

Line 103 – either here or when it's first introduced it would be useful to give a sentence or two of context
– how does this collector and the operation carried out compare to the scale of an actual possible mining
operation- this currently only appears in the Discussion.

**Answer**

We have revised the Introduction accordingly.

Lines 57-60: "The Patania II nodule collector of the Belgian contractor Global Sea Mineral Resources NV
(GSR) is smaller in size and weight than an industrial-scale prototype collector and was tested without a
riser system^{2,17}. However, it does integrate caterpillar tracks for locomotion and hydraulic suction for nodule
collection at great water depth, a technological scheme widely considered feasible for future mining
activities¹⁸. "

Line 103 – 50 m and 4-m the "-" seems to be used variably

**Answer**

Corrected. – Lines 95–96.

Line 108 – Needs a reference for this – what have other studies shown about topographic control on gravity
currents?

**Answer**

The sentence has been removed to limit the manuscript size. The topographic control on gravity currents
based on other studies has been commented on a previous remark about of Reviewer #3.

141 – Delete 'before Detail

**Answer**

Corrected.

152 – I think you need to say "measured background level" if there are no long-term measurements

**Answer**

Corrected. The word "measured" has been added. – Line 141

Line 165 – At what distance were these observed?

**Answer**

The sediment ripples were observed from the last mining lane up to 150 m from the 1st and 3rd mining
strips (end of ortho-photomosaic).

Line 170 – I'd also state that this is the downslope direction in this area.

**Answer**

Added. Lines 165-166: "Larger sediment ripples were mapped at the NE end of the first mining strip
(downslope direction), where the gravity current focused into the steeper NE running valley (Fig. 4)."

Line 202 (and throughout) – sometimes en-dashes with spaces are used to separate numbers and other
905 times not, this should be consistent

**Answer**

The whole manuscript has been updated accordingly. The hyphens between numbers and dates have been
replaced with an en-dash without space between numbers or dates.

Line 230 – these videos are great!

**Answer**

We appreciate this positive comment.

Line 269 – what is the sensitivity of the PartiCam for detecting fine material?

**Answer**

The minimum particle size that can be detected with this optical setup is 0.0153 mm with a pixel resolution
of 0.0051 mm. It is now written in lines 307–308 and lines 839–840 of the revised manuscript.

Line 274 – How does this compare with e.g. Spearman et al. 2018?

**Answer**

We have reworked the PartiCam data. The D_{50} particle size was $\approx 110 \mu\text{m}$ for the first mining episode and \approx
$150 \mu\text{m}$ for the second and third episode (please see our detailed response to the seventh comment of
Reviewer #2). Spearman et al. 2018 showed also in-situ sediment flocculation, which reduced the plume
extent. However, they did not provide the in-situ sediment size distribution (e.g., $D_{25} - D_{75}$) rather than
specific intervals: Microflocs $< 160 \mu\text{m}$ and Macroflocs $> 160 \mu\text{m}$. Thus, a direct quantitative comparison is
not possible.

Line 286 – Could this also be quantified with image-derived DEMs? There will be complicated drape over
that terrain so the actual average thicknesses could be determined?

**Answer**

This is an excellent comment that we worked in depth. Three different approaches were applied:

1) The subtraction of the post-impact DEM from the pre-impact DEM is not straightforward due to the
offsets between the two datasets. While the horizontal offsets can be corrected with georeferencing in XY
direction, the vertical offsets are difficult to resolve at a mm level needed for such an estimation. We should
remind here that the image-derived DEMs have a lower spatial resolution (5 mm) than ortho-photomosaics
(2 mm). To increase the image-derived depth map resolution (e.g., down to 2 mm) different camera systems
should be used. Nevertheless, two other approaches worked well within the collector impact site. To show
the applicability of those methods, we picked up one of the seafloor parts, where the maximum sediment
redeposition is expected. This is located within the second mining strip, and it is a flat ($< 1^\circ$), narrow (8 m
width), unmined seafloor part between mining lanes (Fig. 10 in revised manuscript). Across this seafloor
corridor, several turbidity currents from both directions have propagated and redeposited sediment.

2) The image-derived height of the exposed part of individual nodules (based on the pre-impact image-
derived DEM) agrees with the available ground-truth data, showing that most nodules have an exposed
height of 1-3 cm. This is now shown clearly in Figure 9 of revised manuscript. As these nodules are fully
covered by redeposited sediment, the sediment redeposition was ≈ 3 cm in this location.

3) The presence of polymetallic nodules embedded on the sediment surface alternates seafloor roughness
(compared to the same seafloor without polymetallic nodules or substantially redeposited sediment). Thick
sediment redeposition smooths out this roughness as it fills the gaps between neighbouring nodules,
covering the nodules humps or the nodules entirely. The degree of seafloor roughness at this cm-scale
needed was quantified by using the Terrain Ruggedness index (TRI) in the image-derived DEMs before and
after the mining trial. Outside the collector impact site, analysis at successive distances (50 to 2000 m)
showed that this method could not be applied. While sediment redeposition exists, the sediment thickness
is not enough to alter the nodules' ruggedness. In addition, the image-derived DEMs produced by a single
survey line (with a single camera) have lower depth quality as multiple acquisition angles are needed for
high-resolution image-derived depth maps.

We have updated the revised manuscript accordingly:

Lines 315–328: "As the height of the nodules above the sediment surface at this site is 1–3 cm^{17,50}, the
sediment redeposition was at least 3 cm in these locations (Fig. 9 and Fig. S14). The redeposited sediment
smoothed out the convex shape of nodules and filled the gaps between neighboring nodules, resulting in
decreased seafloor ruggedness (Fig. 9)."

**Fig. 9 Sediment redeposition of up to 3 cm was measured within the mining site.**

**a)** Seafloor orthophoto-mosaic before the mining trial within the impact site. Exact location of Fig.9 is shown in Fig. 10. a1)
 Hill shade of the same seafloor part based on the image-derived DEM. a2) Terrain Ruggedness Index (TRI) of the image-
 derived DEM (see Methods). **b–b2)** Orthophoto-mosaic, hill shade, TRI of the same seafloor part after the mining trial. The
 sediment redeposition has covered the nodules entirely (b) and smoothed out the seafloor relief (cm scale; b1 and b2). **c)**
 Image-derived DEM of the seafloor before the mining trial showing large nodules. **d)** The height profile over a large nodule
 shows that its exposed part reaches almost 3 cm above the seafloor surface, and it is entirely covered with redeposited
 sediment (b).

Line 298 – What is this computer-based analysis? I'd add the name even though it's in the methods.

**Answer**

The term computer-based analysis describes here the use of software (Mangan Analyzer) to automatically
 count the % seafloor nodule coverage in AUV seafloor images before (≈ 29 h) and after (≈ 72 h) the mining
 trial. The sentence has been updated accordingly.

Lines 330–331: "A computer-based quantitative analysis of seafloor images with the Mangan Analyzer
 software (see Methods) showed".

Line 322 - I think somewhere in the Discussion it would be useful to clarify if this density current behaves
 in a similar way to others measured on the seafloor. Maybe in the section where the authors discuss
 concentration decay.

**Answer**

This was also commented on response to a comment about line 22 from Reviewer #3 (please see there).

Line 361 – I think particle size is important here too?

**Answer**
We agree that particle size and particle size distribution (in relation to MBES frequency used) play a major
role in detecting and quantifying sediment concentration by water-column MBES. The current publicly
available studies have used MBES up to 455 kHz⁶⁷⁻⁶⁸. The results from these studies show that this
methodology can be used for larger particle sizes, flocculants and relatively high sediment concentrations.
This is the reason that we have already included the phrase *at "least for larger aggregates"* – line 407 in
revised manuscript.

Line 368 – or higher? Or the integration of ADCPs?
**Answer**
Regarding higher frequencies, the current commercially available nominal MBES frequencies are up to 1400
989 kHz, three times larger than the frequency used here. To the best of our knowledge, this MBES is not yet
depth-rated down to 6000 m and has not been integrated into deep-sea AUVs or seafloor observatories.
This could be possible upon special request to the MBES and AUV manufacturers. We hope that this study
will initiate a development in this direction, as the use of MBES at 400 kHz showed adequate results and
potential for further research and technical improvements.
Regarding the integration of ADCPs: ADCPs are currently integrated as DVLs into AUVs like the one used in
this study. However, the ADCP information of the DVL is not stored and the DVL is only used to help the
AUV maintain proper navigation.

Line 373: Used OBS rather than turbidity earlier
**Answer**
Corrected. The term OBS was used here, as it refers to the sensor and not to data. It reads now "...by the
OBS on the AUV." - Line 409 in revised manuscript.

Line 406 – Like that provided by modelling?
**Answer**
The outcome message of lines 402–407 in the old manuscript was that the results of this study, particularly
the suspended sediment concentration decay with distance and altitude, cannot not be projected to other
trials with different initial conditions (e.g. discharge rate) or other nodule collectors, different sediment size
distribution, seafloor morphology and bottom current regime. Nevertheless, these high-quality in-situ
measurements could enrich existing or developing 4D plume dispersion models, in which different
scenarios could be examined. This outcome becomes clearer in the revised version.
Lines 431–436: "The fate of the plume, as it spreads over longer distances and time spans, and the thickness
of the redeposited sediment can currently only be derived from 4D numerical models. Improved by in-situ
data such as those presented here and accounting for particle flocculation²³ and seafloor morphology, such
models can provide reliable forecasts under different initial conditions and taking account of sporadic
environmental events, such as mesoscale eddies reaching the seafloor⁷²⁻⁷⁴."

Line 409 – In this case, but in future experiments those could be measured.
**Answer**
We agree that tracking the plume over longer distances and time spans could be possible. A combination
of long-ranging AUVs, seafloor observatories and full ocean depth moorings could contribute. This is why
we have included the word currently in our initial sentence. - line 432 in revised manuscript.

Line 413 – Maybe reference Gillard, B. et al. Physical and hydrodynamic properties of deep sea mining-
generated, abyssal sediment plumes in the Clarion Clipperton Fracture Zone (eastern-central Pacific), Elem.
Sci. Anth. 7, 5. <https://doi.org/10.1525/elementa.343>. Here?
**Answer**
The reference has been added. - line 434 in revised manuscript.

Line 424 – will the resuspension of sediment draped by other tracks potentially increase the sediment
discharge rate or do the authors think this effect will be insignificant?

**Answer**

A near-field plume study (< 120 m from the collector impact site) showed that the collector's movement
resuspended 3 ± 2 kg s⁻¹ of sediment when all pump heads were turned off²⁰. This resuspended sediment
is not entrained into the nodule collector; thus, it does not increase the sediment discharge rate. However,
it could contribute to the total plume concentration within the impact site. The nodule collector picks up
plenty of freshly resettled material as well. The resuspension of freshly resettled material could explain the
higher concentration measured in this studied than described in MIT study. (Please also see the 10th and
11th general remark from Reviewer #2). It is still unknown if the freshly resettled material is integrated into
the gravity flow and far-field plume transportation. We expect that the resuspension of sediment draped
by other tracks, particularly when one or more nodule collectors move continuously for weeks or months
over the seafloor, could have a considerable role in environmental impact. However, this resuspended
sediment plume has different characteristics (e.g., sediment size distribution, settling velocity) and
behaviour than the primary discharge plume. Detailed studies should be done in this direction. We added
the following text:

Lines 440–447: " Additionally, the sediment in suspension around the collector heads and caterpillar tracks
should be considered. This resuspended sediment is not entrained into the nodule collector, having
different characteristics (e.g., sediment size distribution) and behaviour than the discharge plume. It
contributes to the total plume concentration within the impact site, but it is not clear if it is integrated into
the gravity current and far-field plume transportation. Moreover, the merging of multiple gravity currents
generated by the mining vehicle(s) is a field that should be investigated in detail⁷⁵. Both could be of interest
if one or more nodule collectors operate continuously for weeks or months over the seafloor. "

Line 440 – I agree and I think you could do even more with this incredible data!

**Answer**

We welcome this very positive comment. Indeed, the analysis of imagery acquired extends beyond this
manuscript. We are working on a manuscript dedicated to the ortho-photomosaics before and after the
mining trial.

Line 456 – Not sure if the switch to turbidity in this section is deliberate but it would be an OBS sensor

**Answer**

We switched to the term turbidity data as we refer to data communication. OBS (Optical Backscatter
Sensors) are sensors, while turbidity is one of the OBS output data. – line 481 in the revised manuscript.

Line 473 – Lose "dire"

**Answer**

Corrected. It has been deleted.

Line 475 – "Race to the bottom" needs rewording

**Answer**

Corrected. The new text is:

Lines 496–499: "This in-situ study is part of a worldwide effort to increase our understanding of deep-sea
mining and the requirements that future monitoring technologies should meet at a time that the
International Seabed Authority is drafting regulations for the exploitation of deep-sea minerals⁶, and
several nodule collectors are under development¹⁸."

Line 508 – Reference format

**Answer**

Corrected. Now in line 513.

Line 589 – Reference format

**Answer**

Corrected. Now in lines 606–607.

Line 624 – Reference format

**Answer**

Corrected. Now in line 654.

Line 719 – Say how far away from the study region this is

**Answer**

This was also commented in comment 4 of Reviewer #3 (please see our response there).

Line 761 – Aha! Good this is known, I'd add this in response to my earlier comment.

**Answer**

It has been added.

Line 780 – considered -> assumed

**Answer**

Corrected. Now in line 862.

Figure comments:

Figure 1: (a) looks like a screen grab. Resolution needs improving and the text is hard to make out. I would
make this larger and higher resolution, improve the enlarged map panel to include lat/lons and a proper
scale (the numbers mean nothing to most people); (c) also looks a bit pixelated but maybe that is my screen.
I also question the usefulness of a single profile that is necessarily vertically exaggerated. It might be more
useful to show multiple profiles or none at all – the need to compress the horizontal scale makes the slopes
look far more extreme than they are; (d) The NW-SE artefacts are not all >7 degrees as stated in the caption,
some appear to be 5-7. Also, I think these are maps made in QGIS – as this is an open-source software it
should be referenced.

**Answer**

Following the abovementioned recommendations, we have increased the resolution and text font size (a).
We have enlarged the map panel and added Lat/Lon and a scale bar (b). We removed the single profile
(former c), as it is also shown in Fig. 5. d) The MBES artefacts along the NW–SE direction create artificial
slopes > 6.5 degrees. A better color classification was applied. The revised figure and caption are given in
lines 109–121 of the revised manuscript.

As regards QGIS, we have added the following sentence in Methods section (lines 736–738): "...calculated
using the Riley formula⁸⁰ as it was applied in the free and open-source software QGIS v3.24, which was also
used to create all maps presented here."

Figure 2: I think all these images are amazing, but it would be useful to have lat/lons on them. The rectangle
on (a) drawn for the image in (b) is a different shape to the image shown. The scales for b2 and c2 need
more explanation - these initially look like difference maps but I don't think they are. It is unclear what "top
of the redeposited sediment" means in this context – I think they mean the redeposited piles of nodules?
This needs to be clearer. Was there imagery collected before the survey/in undisturbed regions? This would
be a GREAT comparison to show the actual impact on the seafloor. (b3) one of the grid lines doesn't reach
the top of the graph, and there should be a a/a' on b2 to show the direction of the profile.

**Answer**

We have added Lat/Lon coordinates in the main image (a). We have not added coordinates in the inlet
images to increase the available space for plotting. The position of all inlet images is shown in image a. The
rectangle in (a) drawn for the image in (b) has been corrected. The horizontal and vertical scale for image B
(b1-b3) is given in b3 as a profile. Images c1 and c2 have independent horizontal (and vertical-only for c2)

scales. The bathymetric profile is taken based on the image-derived DEM after the end of the mining trial.
Due to redeposited sediment and sediment compaction under the caterpillar tracks, the seafloor elevation
may have been altered (mm to cm scale within the collector impact site). The zero-depth refers to the
unmined seafloor between the two caterpillar tracks along the same mining lane. The grid line height within
the bathymetric profile has been corrected, and a d / d' has been added to show the direction of the profile.
The revised figure and caption are given in lines 122–137 of the revised manuscript.

Yes, we have collected imagery before and after the mining trial. We are working on a manuscript dedicated
to ortho-photomosaics and image-derived DEMs before and after the in-situ trial. The comprehensive
analysis of the ortho-photomosaics is beyond the aims of this manuscript, which focuses on plume
dispersion. Nevertheless, we have added a small ortho-photomosaic part (b1) that shows the intact seafloor
(line 122). We have also added the Figure 4 (lines 219–226), Figure 9 (lines 320–328) and Supplementary
Figure 16 (lines 108–116 in the revised Supplementary Information), which show the seafloor before and
after the passing of the gravity current.

Figure 3: (b1, b2) what is the approximate field of view here? (c, d) How long after the survey were these
images taken? (f) Can't really read the text on the rose diagram; (g) Resolution on this is bad again although
this could be the review copy?

**Answer**

b1, b2) Since there are no platforms with known dimensions on the seafloor or ROV laser points, we can be
based only on the width of mining lanes (2 m each) and sensors deployed. We are providing a similar video
frame grab with a sensor of known dimensions: ≈ 80 cm height x 60 cm width. Please see our response and
figure to comment #2 from Reviewer #2. We also decided to include this image within Figure 3. c1, c2)
These images were taken ≈ 87 –110 hours after the mining trial. f) We have increased the figure size, font,
and resolution (g). The revised figure and caption are given in lines 201–218 of the revised manuscript.

Figure 4: Font sizes too small. (a) this is a slightly strange way of displaying ADCP data – it could do with a
bit more explanation for what is actually shown, is it clipped? I am not an ADCP expert so other reviewers
may have more comments on this. I would expand the timescale on (c) so we can make out the details,
especially given there is a period of no data shown.

**Answer**

We have increased the font size. a) The data are clipped in the x axis (time) but not in y axis (altitude). The
2 MHz ADCP has limited vertical range. We have revised the figure caption. b & c) We have expanded the
timescale. The revised figure (now as Fig. 7) and caption are given in lines 259–270 of the revised
manuscript.

Figure 5: Why does the scale stop at 0 on (a)? What is the sensitivity of the PartiCam? From this it looks
like maybe 20 μm ? I see that this is in the methods but I'd put it here too.

**Answer**

Figure 5 has been reworked. Please see our response in comment #7 from Reviewer #2. The revised figure
and caption (now as Figure 8) are given in lines 299–310 of the revised manuscript.

Figure 6: (a) Could you timestamp the bottom current track? Also, is it not possible to be more quantitative
in your measurements of sediment cover using image-derived DEMs like you did for the tracks? In (b) the
nodule coverage dots could be enlarged so they're easier to make out. (b1-5) the caption says this is shown
pre-and post but these images are all post? I think this refers to the lines in (b) but images would be nice.
Also are these labelled the wrong way around in (b)? The line labelled "pre" appears to have thicker
coverage than post? But maybe I am misreading them because they're so thin? I'd also make the colours
used for 2-10 and 30-37% more different.

**Answer**

Based on the abovementioned recommendations, we have timestamped one of the bottom tracks (we
think that timestamping all bottom current tracks will overload the figure) and enlarged the nodule dots.
The use of image-derived DEM as a quantitative measure for sediment redeposition is now included (lines
320–328). The caption b refers to pre and post-survey lines. The labels pre and post are correct as larger %
nodule coverage implies less sediment redeposition (please see our answer in comment #6 from Reviewer
#1). The colors for 2-10 % and 30-37 % are different (dark blue and dark red, respectively) and not shades
of the same color. The revised figure (now as Fig. 10) is in lines 359–368.
We have added an additional figure (Fig. S16; lines 108–116), where the seafloor is shown before and after
the sediment redeposition. Please see our response in comment #6 from Reviewer #1.

ED3: can some idea of scale be given? I know this is hard in 3D.

**Answer**

The length of plume based on MBES WCI records and bathymetric profile are provided. The revised figure
(now as Fig. 5) and caption are given in lines 227–233.

ED7: Give location

**Answer**

The location of Extended Data Fig. 7 (now as Supplementary Figure 14) is noted within the legend of Fig. 10
– lines 359–368.

ED8: What is the field of view?

**Answer**

The distance between the legs of the GMR_PFM-40 tripod is 0.75 m. A scale has been added. The resolution
of image Extended Data Fig. 8b does not allow a similar calculation based on the NIOZ_PFM-04 tripod. The
revised figure (now as Supplementary Figure 15) and caption are given in lines 99–107 of the revised
Supplementary Information.

ED9: Scale?

**Answer**

The cross-sectional area has a length of 640 m and acts as a horizontal scale. It is now written with bold
font and larger font size to be more apparent. The revised figure (now as Supplementary Figure 17) and
caption are given in lines 117–123 of the revised Supplementary Information.